# Neuronal dynamics of cerebellum and medial prefrontal cortex in adaptive motor timing

Zhong Ren[1], Xiaolu Wang [1], Milen Angelov[1], Chris I. De Zeeuw [1,2] & Zhenyu Gao [1,3] ✉

Precise temporal control of sensorimotor coordination and adaptation is a fundamental basis of animal behavior. How different brain regions are involved in regulating the flexible temporal adaptation remains elusive. Here, we investigated the neuronal dynamics of the cerebellar interposed nucleus (IpN) and the medial prefrontal cortex (mPFC) neurons during temporal adaptation between delay eyeblink conditioning (DEC) and trace eyeblink conditioning (TEC). When mice were trained for either DEC or TEC and subsequently subjected to a new paradigm, their conditioned responses (CRs) adapted virtually instantaneously. Changes in the activity of the IpN neurons related to CR timing were prominent during DEC-to-TEC adaptation, but less so during TEC-to-DEC adaptation. In contrast, mPFC neurons could rapidly alter their modulation patterns during both adaptation paradigms. Accordingly, silencing the mPFC completely blocked the adaptation of CR timing. These results illustrate how cerebral and cerebellar mechanisms may play different roles during adaptive control of associative motor timing.

Efficient coordination of daily movements often requires establishing associations with specific sensory inputs. Timing plays a crucial role in optimizing associative sensorimotor behavior: the movement must occur at a precise moment following the sensory cue[1]. For instance, a predator needs to launch itself with precise timing to be able to catch its prey. To enhance the temporal alignment between sensory perception and desired motor responses, brain circuits must adapt the temporal aspects of sensorimotor associations[2–5]. Understanding how the brain learns and produces temporally precise movements while maintaining flexible reaction timing remains challenging in neuroscience.

Pavlovian eyeblink conditioning (EBC) offers an ideal experimental model to explore the precise temporal regulation of associative sensorimotor behaviors[6–9]. In EBC training, an innocuous conditioned sensory stimulus (CS) is paired with an aversive unconditioned stimulus (US), which elicits an unconditioned eyeblink response (UR). Over repetitive CS-US pairing, a CS-induced conditioned response (CR) gradually develops, facilitating a well-timed transformation from sensory input to motor output[6,8–12]. The timing of the CR is critically dependent on the temporal relationship between the CS and US. In delay eyeblink conditioning (DEC), where the US occurs at the end of the CS, the CR will occur at the late phase of the CS epoch[10–12]. In trace eyeblink conditioning (TEC), the US is presented after a stimulus-free interval following the CS termination, leading to a CR within this interval[6–8]. Thus, EBC training with distinct CS-US temporal patterns allows an identical sensory input to prompt movements with differing timings.

The cerebellum is a crucial region for the acquisition and expression of CRs during EBC[13–17]. The mossy fiber and climbing fiber pathways transmit CS- and US-related inputs, converging onto the Purkinje cells in the cerebellar cortex[10,14]. By repetitively pairing these two inputs with a consistent interval, well-timed neural modulation can be established in the cerebellar interposed nucleus (IpN), ultimately driving eyelid closure[10,14,18–22]. Recent studies revealed both CS- and US-related activities in Purkinje cells and IpN neurons during DEC and TEC[6,9,10,23]. Stimulating IpN neurons directly triggers eyelid closure,

[1]Department of Neuroscience, Erasmus MC, Westzeedijk 353, 3015 AA Rotterdam, the Netherlands. [2]Netherlands Institute for Neuroscience, Royal Dutch Academy of Arts & Science, 1105 BA Amsterdam, the Netherlands. [3]Department of Neurosurgery, Erasmus MC, Westzeedijk 353, 3015 AA Rotterdam, the Netherlands. ✉e-mail: z.gao@erasmusmc.nl

whereas inhibiting IpN outputs significantly hampers both the CR acquisition and expression[8,9,11,12,21,22,24–26]. This underscores the causal relationship between IpN activity and the CR performance. Nonetheless, it remains unclear whether, once the CRs are established, the temporal features of these learned movements can still be adaptively modified. And if so, what are the specific brain regions that mediate the adaptation?

The medial prefrontal cortex (mPFC) is a higher-order association cortex crucial for adaptive control of sensorimotor tasks, and is therefore likely to contribute to the flexible regulation of sensorimotor timing[27,28]. Specifically, the mPFC plays a permissive role in gating TEC, but not DEC, highlighting its importance for the temporal features of learned movements[14,16,29–36]. It has been suggested that mPFC neurons transmit CS-related signals through the pontine nuclei to the cerebellum and establish a short-term memory trace that permits the CS-US association during TEC[33,35,37,38]. Therefore, we hypothesize that both the cerebellum and mPFC are required for flexible control of sensorimotor timing, yet with differential impacts.

To test this hypothesis, we investigated the shared and distinct neural dynamics of the IpN and mPFC neurons during DEC and TEC. We focused specifically on understanding how mice adjust their CR timing in response to the unexpected CS-US intervals. We found that mice exhibited almost instantaneous adjustments of CR onset timing when subjected to adaptation from DEC-to-TEC or TEC-to-DEC paradigms. Remarkably, the temporal adaptation of the CR was reflected in IpN neural activity during the DEC-to-TEC switch, but less so during TEC-to-DEC adaptation. In contrast, mPFC neurons had consistent alterations of their task-related and baseline activity patterns during the paradigm switches. These findings were further supported by our perturbation experiments in that suppressing mPFC activity hindered animals' ability to swiftly adapt CR timing during both DEC-to-TEC and TEC-to-DEC transitions. In summary, our results provide new insights into the cerebral and cerebellar mechanisms that underlie adaptive temporal control of learned associative movements.

## Results

### Distinct CR kinematics in mice trained with DEC and TEC paradigms

We first trained two separate cohorts of mice, using either the TEC or DEC paradigm, for a direct comparison of their behavioral and neuronal characteristics. Both TEC and DEC trials began with a 250 ms flashlight stimulus serving as the CS, paired with a 15-ms air-puff to the left eye as the US. In the DEC paradigm, the US followed the CS immediately, while a 250 ms interval was placed between the CS offset and the US onset in the TEC paradigm (Fig. 1a, c). Our experimental design aims to investigate how identical sensory inputs, presented at different CS-US intervals, lead to distinct temporal input–output features in behavior after sensorimotor learning[8,10–12].

Mice developed specific temporal features of their CRs following TEC and DEC training. TEC-trained mice had later CR onset and peak time compared to those of the DEC-trained mice (onset time: $300.38 \pm 5.77$ ms in TEC vs. $184.26 \pm 2.56$ ms in DEC, $P = 5.73 \times 10^{-25}$; peak time: $446.40 \pm 2.30$ ms in TEC vs. $235.88 \pm 1.01$ ms in DEC, $P = 6.0 \times 10^{-27}$; $n = 92$ and 67 sessions, Fig. 1d, e). The intervals between the CR peak time and the US onset were $27.37 \pm 6.61$ ms for the TEC group and $50.87 \pm 7.02$ ms for the DEC group ($P = 0.08$, $n = 92$ and 27 sessions). In addition, TEC-trained mice showed shallower CR slopes and lower CR peak velocities compared to the DEC-trained mice (CR slope: $0.20 \pm 0.008$ in TEC vs. $0.53 \pm 0.026$ in DEC, a.u., $P = 1.0 \times 10^{-21}$; CR peak velocity: $6.7 \pm 0.25$ in TEC vs. $10.3 \pm 0.54$ in DEC, a.u., $P = 7.7 \times 10^{-10}$, $n = 92$ and 67 sessions, Fig. 1d, f, g). The CR amplitudes of TEC- and DEC-trained mice were comparable ($40.23 \pm 1.59\%$ in TEC vs. $40.57 \pm 2.00\%$ in DEC, $P = 0.85$, $n = 92$ and 67 sessions, Fig. 1h). To rule out the possibility that distinct CR kinematics in TEC- and DEC-trained mice were influenced by specific

detection method, we employed three well-accepted methods to determine the precise CR onset timing[11,39]. All three detection methods consistently illustrated the fundamental difference in CR timing in DEC and TEC training mice (Fig. 1e, Supplementary Fig. 1a–d). These findings indicate that, despite the use of identical CS stimuli, the temporal specificity of the CS-US intervals determines specific CR timing and kinematics, which ensures the optimal conditioned eyelid closure before US onset.

We recorded the activity of cerebellar interposed nucleus (IpN) neurons ipsilateral to the trained eye during TEC and DEC (Fig. 1b). Among TEC-trained mice, a total of 138 IpN neurons exhibited CR-related facilitation (Fig. 1i, Supplementary Fig. 1e), with a peak increase of $52.83 \pm 2.48$ Hz (mean ± s.e.m., Fig. 1k). For DEC-trained mice, we recorded 51 IpN neurons with facilitation, showing a peak increase of $80.71 \pm 5.75$ Hz (mean ± s.e.m., Fig. 1j, k, Supplementary Fig. 1e). In addition, we detected 71 suppression IpN neurons (with a suppression of $37.12 \pm 1.75$ Hz, mean ± s.e.m.) from TEC-trained mice (Fig. 1l, n, Supplementary Fig. 1f), and 27 IpN neurons (with a suppression of $45.92 \pm 2.24$ Hz, mean ± s.e.m.) from DEC-trained mice (Fig. 1m, n, Supplementary Fig. 1f). Despite the similarity in CR amplitudes between the two cohorts (Fig. 1h), the IpN modulation was notably larger in DEC-trained mice (Fig. 1k, n, facilitation groups: $P = 4.4 \times 10^{-7}$, $n = 138$ and 51 cells; suppression groups: $P = 0.0008$, $n = 71$ and 27 cells). These results unveil the task-related neuronal dynamics within IpN, possibly reflecting a connection between the strength of IpN neuronal modulation and CR kinematics.

### Temporal relationship between IpN activity and CR

CS-triggered IpN facilitation activates the downstream premotor nuclei, which consequently drives CRs in both TEC and DEC[11,14,16,19,40]. To what extent do the temporal patterns of IpN neurons encode the temporal feature of CRs in DEC and TEC paradigms? We analyzed the temporal relationship between IpN modulation and CRs in DEC- and TEC-trained mice (Fig. 2a). The facilitation onset of IpN neurons consistently preceded CR onset in both TEC and DEC groups (Fig. 2b, Supplementary Fig. 2a, b, TEC facilitation onset vs. CR onset: $138.18 \pm 8.29$ ms vs. $295.59 \pm 4.65$ ms, $P = 5.5 \times 10^{-15}$, $n = 138$ cells. DEC facilitation onset vs. CR onset: $105.10 \pm 5.84$ ms vs. $189.60 \pm 2.92$ ms, $P = 2.0 \times 10^{-15}$, $n = 51$ cells). Despite the evident disparity in CR timings (Fig. 2b, right distribution histogram, $P = 1.2 \times 10^{-23}$), the onset timings of IpN modulation remained comparable between the TEC and DEC groups (Fig. 2b, top, $P = 0.34$). Consequently, a larger interval (ΔOnset) between IpN modulation and CR onset was observed in the TEC group (TEC Δonset: $157.41 \pm 9.77$ ms, DEC Δonset: $84.50 \pm 6.40$ ms, $P = 1.58 \times 10^{-9}$). To evaluate if IpN neurons encoded CR onset timing on a trial-by-trial basis, we conducted a temporal correlation analysis between modulation onset and CR onset for all CR-related IpN neurons. For a subgroup of IpN neurons from TEC-trained mice, their CR onset consistently aligned with the modulation onset on a trial-by-trial basis (Fig. 2c, correlation r = 0.72, $P = 8.72 \times 10^{-6}$ for the example neuron, $n = 19$ cells). Similar correlations were also found in the DEC-trained mice (Fig. 2d, correlation $r = 0.77$, $P = 1.04 \times 10^{-5}$ for the example cell, $n = 7$ cells). Hence, a subset of IpN neurons precisely predicts CR onset timing for both TEC and DEC mice.

Previous studies have defined the 'eyeblink-related IpN neurons' as the neurons whose spike rates correlate with the CR amplitudes on a trial-by-trial basis[11,12]. We then compared the timing of CR-related modulation in these eyeblink-related IpN neurons. Consistent with the findings from the population of all IpN neurons, the modulation onset of eyeblink-related IpN neurons showed no significant difference between DEC- and TEC-trained mice, resulting in a significantly larger interval between IpN modulation and CR onset in TEC-trained mice (Supplementary Fig. 2c–e). In addition, similar neuronal activity-behavior relationships were observed in the IpN neurons showing CR-related suppression (Supplementary Fig. 2f). These results showed that

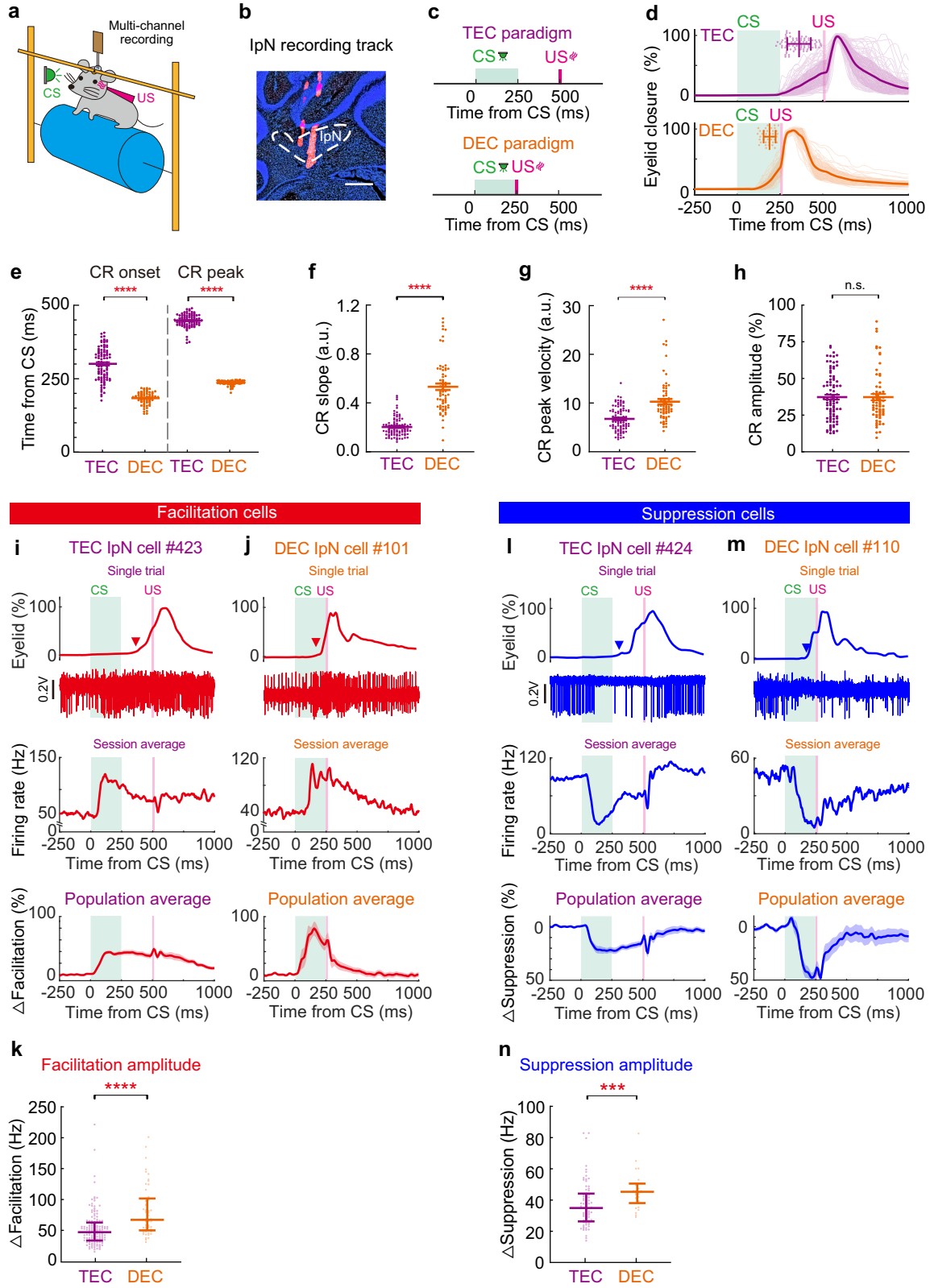

the temporal relationships between IpN output and CR onset were differentially set for the DEC- and TEC-trained mice.

IpN output can directly initiate eyelid closure through a di-synaptic pathway involving the facial nucleus via the red nucleus[12]. We asked whether the interval between IpN modulation and CR onset is determined by the synaptic delay of this pathway. To assess this, we conducted direct electrostimulation of IpN neurons and measured the

minimum induction time required to elicit eyelid closure via the IpN-red nucleus-facial nucleus pathway. With varying current strengths, we observed an increment in eyelid closure amplitudes while maintaining consistent eyelid closure onset timings (Supplementary Fig. 3a). Interestingly, the minimum induction time was significantly shorter than the interval between IpN modulation and CR onset in both DEC- and TEC-trained mice (Supplementary Fig. 3b–d). In addition, IpN

**Fig. 1 | Mice establish well-timed eyelid closure in response to specific CS-US temporal relationships. a** Schematic of the experimental setup. A head-fixed mouse is presented with a green light as the CS and an air puff as the US, with a multichannel silicon probe penetrating into the interposed nuclei (IpN). **b** Example of a DiI-labeled recording track in the IpN (scale bar, 1 mm, one example out of 40 mice recorded in IpN). **c** The CS-US relationships of the TEC and DEC paradigms. **d** Summary of eyelid traces from example TEC ($n = 79$ trials) and DEC ($n = 39$ trials) recording sessions. The scatter plots at the top show the CR onsets of individual trials and the session average. **e–h** Comparison of the CR onset and peak timings (**e**, $P = 5.7 \times 10^{-25}$ and $P = 6.0 \times 10^{-27}$), CR slope (**f**, $P = 1.0 \times 10^{-21}$), CR peak velocity (**g**, $P = 7.7 \times 10^{-10}$), and CR amplitude (**h**, $P = 0.85$) between the TEC- and DEC-trained mice ($n = 92$ and 67 sessions). **i** Neuronal activities of IpN neurons in TEC-trained mice. Top, the eyelid closure curve (triangle indicates the CR onset) and corresponding spike modulation from a single TEC trial. Middle, spike rate modulation of the same IpN neuron during a TEC recording session. Bottom, averaged firing activity of all IpN neurons with spike rate facilitation between CS-US intervals during TEC. **j** Same as (**i**) but for the DEC-trained group. **k** Comparison of the CR-related facilitation of IpN neurons in TEC- and DEC-trained mice ($P = 4.4 \times 10^{-7}$, $n = 138$ and 51 cells). **l, m** Same as (**i–j**), but for the IpN neurons that showed CR-related suppression during TEC (**l**) or DEC (**m**). **n** Comparison of the CR-related suppression of IpN neurons in TEC- and DEC-trained mice ($P = 0.0008$, $n = 71$ and 27 cells). Data are shown as the mean ± s.e.m., except for (**d**), which is the mean ± SD, and for (**k**), (**n**), which are median with interquartile range. Two-sided Mann–Whitney tests are performed in (**e–h**), (**k**), and (**n**). n.s.: not significant, \*\*\*$P \leq 0.001$, and \*\*\*\*$P \leq 0.0001$. Source data are provided as a Source data file.

neurons showed clear CR-related modulation during the trials in which mice did not show CR, although much less pronounced than the trials with CR (Supplementary Fig. 3e, f). Hence, IpN modulation does not exhibit a fixed temporal relationship with CR onset that solely reflects the motor drive for eyelid closure, suggesting that IpN modulation does not encode CR in an all-or-nothing manner. Instead, our data suggests the implementation of multiplex encoding strategies for controlling CR onsets in IpN.

We asked what types of encoding strategies IpN neurons could potentially implement to track the CR onset timing. We trained a decoding classifier to distinguish the early- and late-CR onset trials using the firing dynamics of IpN neurons recorded during TEC[41]. We sorted all the CR trials based on their onset timing and divided them into early- and late-onset trials. Subsequently, a subset of trials from both datasets was selected for training the decoder and the rest of trials were used for assessing the decoding accuracy (Supplementary Fig. 4a; see Methods). The decoding classifier showed that the CR onsets could be decoded based on the modulation timing in a sub-group of IpN neurons (Supplementary Fig. 4b, $n = 11$ cells). Moreover, a distinct subset of IpN neurons increased their firing rates during the late-CR trials (Supplementary Fig. 4c, $n = 22$ cells), while another neurons increased firing rates only during the early-CR trials (Supplementary Fig. 4d, $n = 9$ cells). We also attempted to decode the neuron activity for the CR onsets in DEC-trained mice. However, the decoding results were insufficient due to the minimal variety of CR onset timings during the DEC. Together, our analysis confirmed that IpN neurons employ multiplex encoding strategies to regulate the temporal aspects of CRs. Both temporal and spike rate coding mechanisms can play roles in the trial-by-trial control of CR timing.

## Rapid adaptation of CR onset timing during the DEC-to-TEC adaptation paradigm

Once well-trained mice establish a specific CR onset timing, can they still adjust their CR timing when exposed to an unfamiliar CS-US interval? We introduced the TEC paradigm to the mice previously trained in DEC (Fig. 3a). Mice retained their CR percentages but had significantly smaller CR amplitudes after we switched to the TEC paradigm (Fig. 3b–d, Supplementary Fig. 5a, e). Remarkably, mice rapidly adapted their CR timing to match the novel CS-US interval after the paradigm switch. Both CR onset and peak timings were significantly delayed following the DEC-to-TEC adaptation (Fig. 3b–g, Supplementary Fig. 5b, c, session averaged CR onset in Fig. 3g. DEC: $168.03 \pm 4.28$ ms, TEC: $197.47 \pm 5.64$ ms, $P = 5.0 \times 10^{-5}$, $n = 38$ sessions). The adaptation of CR timing occurred almost instantaneously upon the introduction of the novel TEC paradigm. This rapid behavioral adaptation was consistently observed across different animals (Fig. 3f, Supplementary Fig. 5d, comparing the CR onset of all 4 quarters of TEC epochs with the DEC epoch, $P = 2.72 \times 10^{-13}$, $n = 38$ sessions). Notably the adapted CR timing still differed from that of TEC-trained animals, despite comparable CR amplitudes (Fig. 3g, TEC-trained CR onset: $241.44 \pm 7.61$ ms, $P = 0.0002$, $n = 92$ sessions, Supplementary Fig. 5b, e).

Could the observed shift in CR onset timing during DEC-TEC training be attributed to a gradual extinction of the original DEC response, paired with a simultaneous gradual acquisition of a new TEC response? Using three distinct criteria to determine CR onset, we found that changes in CR amplitude had minimal impact on the timing of CR onset (see Supplementary Fig. 5f, g). To investigate this further, we trained two other cohorts of mice. One cohort underwent DEC training followed by extinction sessions. During extinction, these mice exhibited a gradual reduction in CR amplitude; yet, there was no corresponding delay in either CR onset or CR peak timing (Fig. 3h, $P = 0.23$, $n = 6$ sessions, Supplementary Fig. 6). In the second cohort, DEC-trained mice underwent TEC adaptation with an extended CS-US interstimulus interval, increased from 250 to 500 ms (i.e., DEC-to-TEC750, Supplementary Fig. 7a). This task design was devised to provide a more challenging scenario for the mouse to adapt their CR timing. Our rationale is that if a new TEC response emerges after the switch, it should be readily visible during the CS-US internal, which covers even the original US period in the DEC-to-TEC750 paradigm. Yet, contrary to the possibility that TEC-related CR emerged with a much later timing, the DEC-related CR was not diminished but was gradually shifted to a much later timing, following the DEC-to-TEC750 paradigm switch (Supplementary Fig. 7b, c). These findings collectively indicate that mice can rapidly adjust CR onset timing, a mechanism that appears distinct from the extinction-and-acquisition model previously shown in studies using rabbits[6].

How do the cerebellar neurons respond to an altered CS-US interval? One possibility is that IpN neurons alter their modulation patterns to generate adaptive CRs during the DEC-to-TEC adaptation. We focused on IpN neurons exhibiting CR-related facilitation in both DEC and TEC epochs ($n = 40$ cells). Indeed, we observed adaptation in CR-related modulation upon the paradigm switch (Fig. 4a–d). On average, the facilitation amplitude was decreased by 55.3%, and facilitation onset was delayed by 21.5% after the DEC-to-TEC adaptation (Fig. 4b–d, facilitation amplitudes: $164.64 \pm 31.46\%$ in DEC and $106.00 \pm 19.10\%$ in TEC, $P = 0.0003$; facilitation onset: $92.75 \pm 7.13$ ms in DEC, and $112.70 \pm 9.33$ ms in TEC, $P = 0.001$). However, the delayed facilitation onset didn't precisely align with behavioral changes. Adaptation in IpN modulation onset was significantly smaller than the adaptation of CR onset timing, leading to an extended latency between IpN neuronal activity and behavior (ΔOnset) after adaptation (Fig. 4e, f, DEC epoch: $70.16 \pm 7.24$ ms, TEC: $81.80 \pm 8.80$ ms, $P = 0.0053$). Similar phenomena were also observed in IpN neurons exhibiting CR-related suppression (Supplementary Fig. 8). In summary, our data shows that mice can rapidly recalibrate their CR timing in response to a novel CS-US interval. However, these changes in behavior cannot be solely accounted for by the adaptation of neuronal activity in IpN.

## Rapid adaptation of CR onset timing during the TEC-to-DEC adaptation paradigm

We next investigated whether the TEC-trained mice could flexibly adapt their CR timing when exposed to a shorter CS-US interval in a

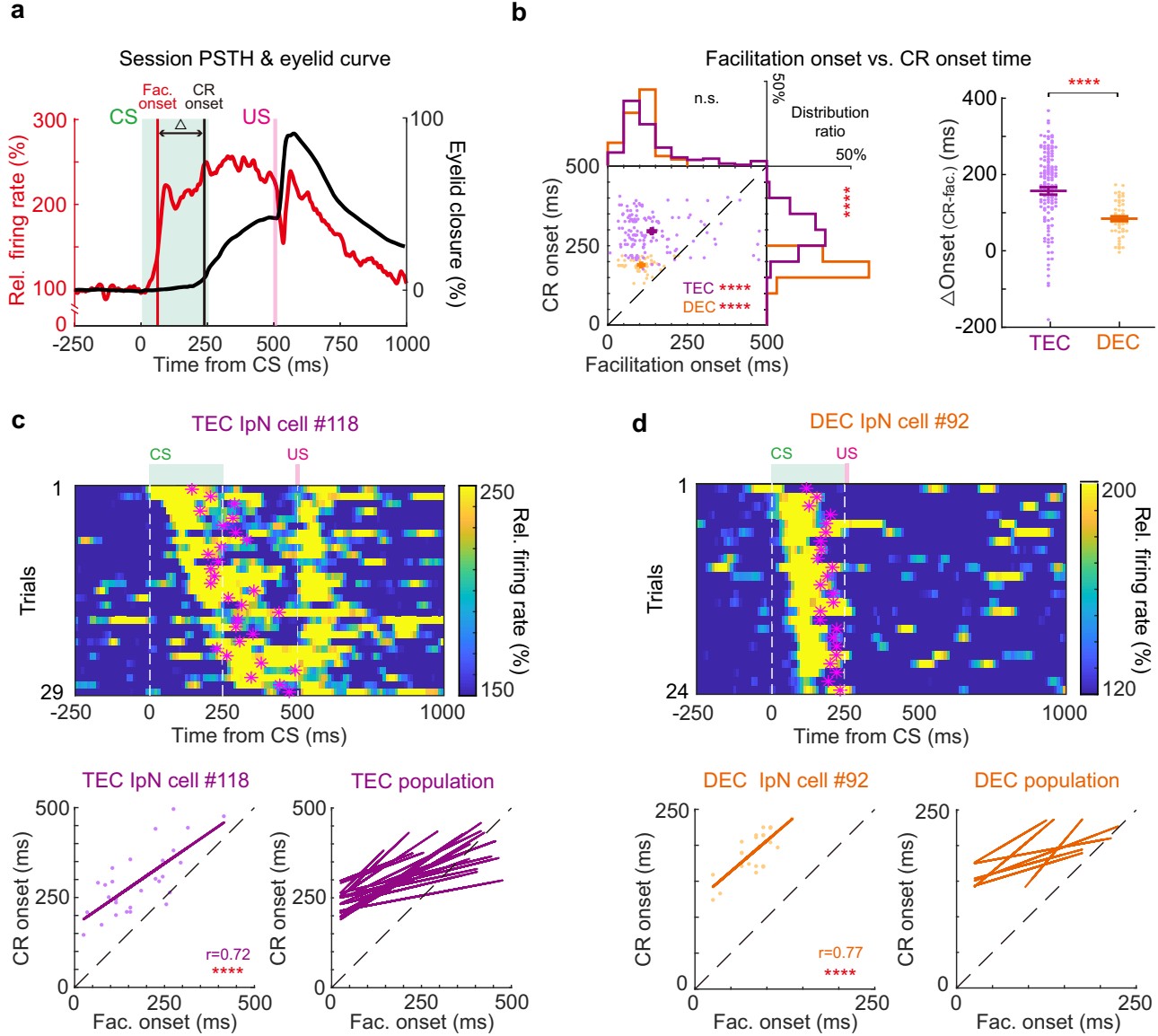

**Fig. 2 | Weak temporal correlation between the IpN population activity and the CRs. a** Illustration of the timing parameters extracted from the IpN neuron spike rate (red) and the eyelid closure (black) traces, including the timing of facilitation onset (Fac. onset) and CR onset, as well as their timing interval (Δ) in between. **b** Left, distribution and histograms of the IpN facilitation onset timing and the CR onset timing from TEC (purple) and DEC (orange) recordings. In both the TEC and DEC groups, IpN facilitation occurred before CR onset (TEC facilitation vs. CR onset, $P = 5.5 \times 10^{-15}$, DEC facilitation vs. CR onset, $P = 2.0 \times 10^{-15}$, two-sided Wilcoxon test). The CR onset distribution in the DEC group was earlier than that in the TEC group ($P = 1.2 \times 10^{-23}$, two-sided Mann–Whitney test), whereas the distribution of IpN facilitation onsets was comparable in these two groups ($P = 0.34$, two-sided Mann–Whitney test, $n = 138$ and 51 cells). Right, the facilitation-to-CR

onset interval (Δ) between TEC and DEC groups ($P = 1.58 \times 10^{-9}$, two-sided Mann–Whitney test, $n = 138$ and 51 cells). **c** IpN cells with significant trial-by-trial correlations between spike modulation onset and CR onset in TEC recordings. Top, heatmap of the instantaneous firing rate from an example IpN neuron during TEC trials, ordered by facilitation onset timing. Magenta stars indicate the CR onsets. Bottom left, facilitation onset-CR onset pairs and their correlation during TEC trials from the example neuron ($r = 0.72$, $P = 8.72 \times 10^{-6}$, linear regression). Bottom right, correlations of all TEC cells ($n = 19$) with this pattern. **d** Similar to **c** but for all facilitation cells ($n = 7$) from the DEC group ($r = 0.77$, $P = 1.04 \times 10^{-5}$, linear regression). Data are shown as the mean ± s.e.m., n.s.: not significant, ****$P \le 0.0001$. Source data are provided as a Source data file.

TEC-to-DEC switch (Fig. 5a). Intriguingly, TEC-trained mice also showed a rapid adaptive response to this paradigm switch, displaying a shortened CR onset timing post-switch (Fig. 5b–g, in Fig. 5g, TEC epoch: 232.30 ± 9.33 ms, DEC epoch: 173.82 ± 4.43 ms, $P = 2.05 \times 10^{-8}$, $n = 33$ sessions). Mirroring the outcome of the DEC-to-TEC adaptation, the TEC-to-DEC adaptation led to nearly instantaneous adjustments in CR onset timing. Within merely 10 trials post-adaptation, CR timings were significantly shortened, with CR kinematics continuing to adapt throughout the whole DEC trials (Fig. 5c–f, Supplementary Fig. 9c, comparing the CR onset of 4 quartiles of the DEC trials with the TEC

trials, $P < 1.0 \times 10^{-15}$, $n = 33$ sessions). Decrease in CR percentage, earlier CR peak time, and increase in CR amplitude were observed during the TEC-to-DEC adaptation (Supplementary Fig. 9a–d). However, these changes in CR kinematics had no impact on CR onset time detection (Supplementary Fig. 9e, f). By the end of the TEC-to-DEC adaptation, CR onset timing and amplitudes became indistinguishable from those mice trained with the DEC paradigm (Fig. 5g, DEC-trained CR onset: 168.02 ± 4.28 ms, $P = 0.14$, $n = 38$ sessions; Supplementary Fig. 9d). Therefore, mice were capable of fully adapting the CR onset timing within a single TEC-to-DEC session.

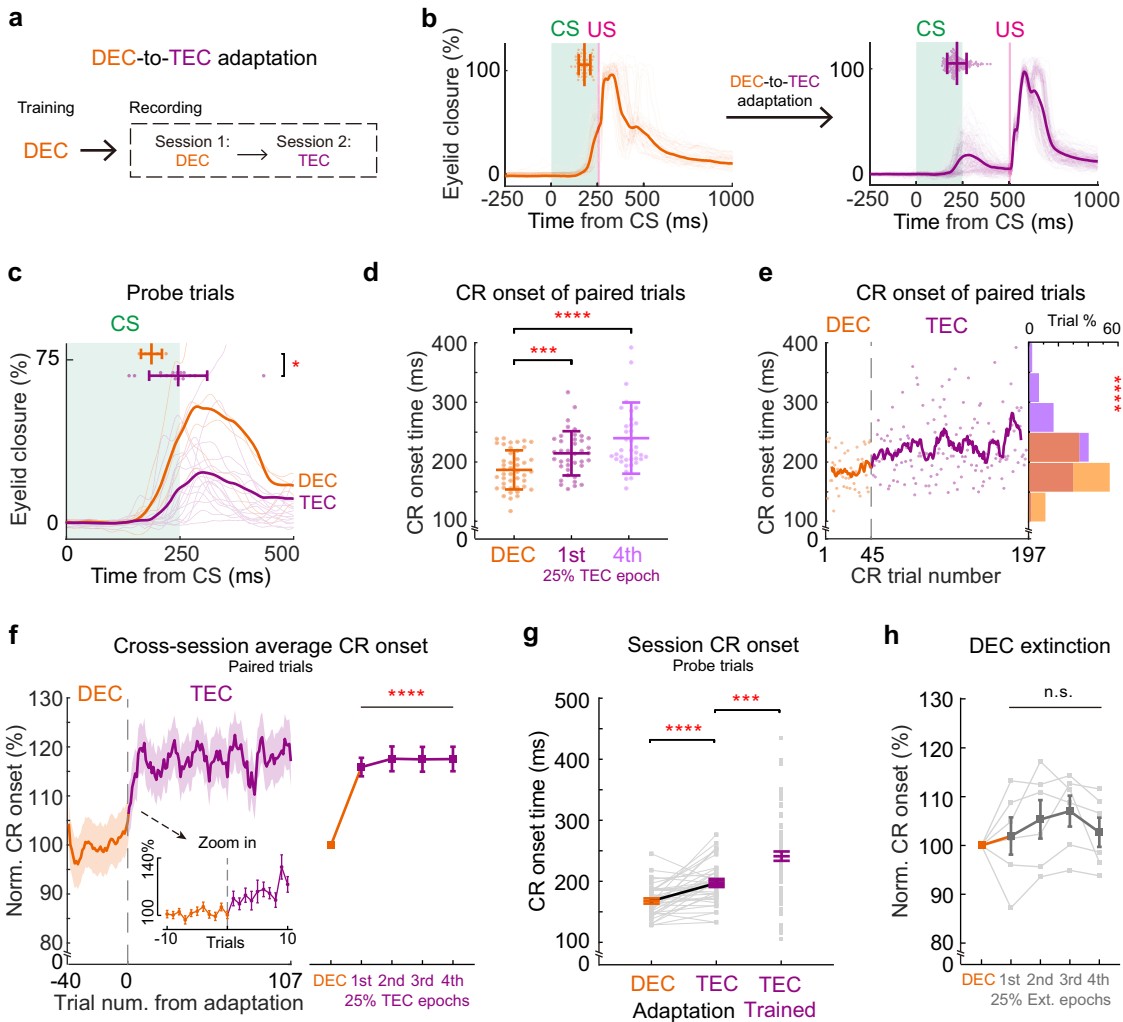

**Fig. 3 | Rapid adaptation of CR onset timing during the DEC-to-TEC adaptation paradigm. a** Experimental procedure for the animal training and paradigm switch from DEC to TEC. **b** Example eyelid closure curves recorded during the DEC (left, $n = 44$ trials) and TEC sessions (right, $n = 153$ trials). The scatter plots at the top show the CR onsets of individual trials and the group averages. **c** Average and individual (in dark and light colors) eyelid closure curves of the DEC (orange) and TEC (purple) probe trials from the session shown in (**b**). The scatter plots at the top show the CR onsets of individual trials ($P = 0.014$, two-sided Mann–Whitney test, $n = 5$ and 17 trials). **d** CR onset timings of all paired trials from the example recording session. The CR onset timings are divided into DEC epoch (orange), the first 25% TEC epoch (purple), and the last 25% TEC epoch (light purple) (DEC vs. first 25% TEC $P = 0.001$, DEC vs. last 25% TEC $P = 2.74 \times 10^{-6}$, first vs. last 25% TEC $P = 0.078$, two-sided Mann–Whitney test, $n = 44$, 39, and 38 trials) **e** CR onsets of each trial during the example recording from DEC to TEC indicating rapid adjustment. The curve illustrates the moving average. The distribution histogram of the CR onsets in DEC

(orange) and TEC (purple) is shown on the right (CR onset time DEC vs. TEC $P = 4.54 \times 10^{-6}$, two-sided Mann–Whitney test, $n = 44$ and 153 trials). **f** Left, cross-session averages of normalized CR onset timing from all DEC-to-TEC adaptation recordings. The inset panel shows the zoom-in data before and after the paradigm switch. Right, statistical analysis of the CR onset timing for the DEC epochs and all four quartiles of TEC adaptation epochs ($P = 2.72 \times 10^{-13}$ in all four comparisons, two-sided repeat measurement one-way ANOVA, $n = 38$ sessions from 11 mice). **g** Comparisons of CR onset timing of all DEC-to-TEC adaptation recordings together with that of the TEC-trained mice ($P = 5.0 \times 10^{-5}$ and 0.0002, two-sided Wilcoxon test and Mann–Whitney test, $n = 38$ and 92 sessions). **h** Normalized CR onset timing from DEC extinction recordings, which are divided into DEC epochs and all four quartiles of extinction epochs ($P = 0.23$, two-sided repeat measurement one-way ANOVA, $n = 6$ sessions). Data are shown as the mean ± s.e.m., except for (**b**–**d**), which are the mean ± SD, n.s.: not significant, *$P \le 0.05$, ***$P \le 0.001$, and ****$P \le 0.0001$. Source data are provided as a Source data file.

Could these changes in behavior be reflected in the activity of IpN neurons? We recorded IpN neurons, focusing on those CR-related facilitation across the TEC-to-DEC adaptation. Despite the swift adaptation of CR timing, we observed no substantial alterations in the modulation patterns of IpN neurons. Both the onset and amplitude of CR-related facilitation persisted unchanged after the TEC-to-DEC adaptation (Fig. 6a, b). Furthermore, pairwise comparisons indicated no statistical differences in facilitation amplitude and onset timing during the adaptation (Fig. 6c, d; facilitation amplitude: $119.19 \pm 17.88\%$ in TEC and $96.33 \pm 7.35\%$ in DEC, $P = 0.40$; facilitation onset: $106.31 \pm 13.39$ ms in TEC and $111.03 \pm 8.90$ ms in DEC, $P = 0.65$, $n = 35$ cells). However, the altered CR onsets without concurrent changes in IpN activity resulted in an even shorter interval between

neuronal activity and behavior (△Onset) during DEC epochs, as exemplified by a specific neuron (Fig. 6e). This change was consistently observed across a broader facilitation neuron population (Fig. 6f; TEC: $129.10 \pm 14.22$ ms, DEC: $64.56 \pm 9.97$ ms, $P = 1.07 \times 10^{-5}$, $n = 35$ cells). The neuronal activity of IpN neurons exhibiting CR-related suppression also remained unaltered during TEC-to-DEC adaptation (Supplementary Fig. 10). Therefore, we did not find significant neuronal adaptation in IpN that could explain the rapid adaptative behavior.

## CR-related activities in mPFC neurons

Our data demonstrates animals' capacity to swiftly adapt their CR timings when introduced to novel CS-US intervals. However, the

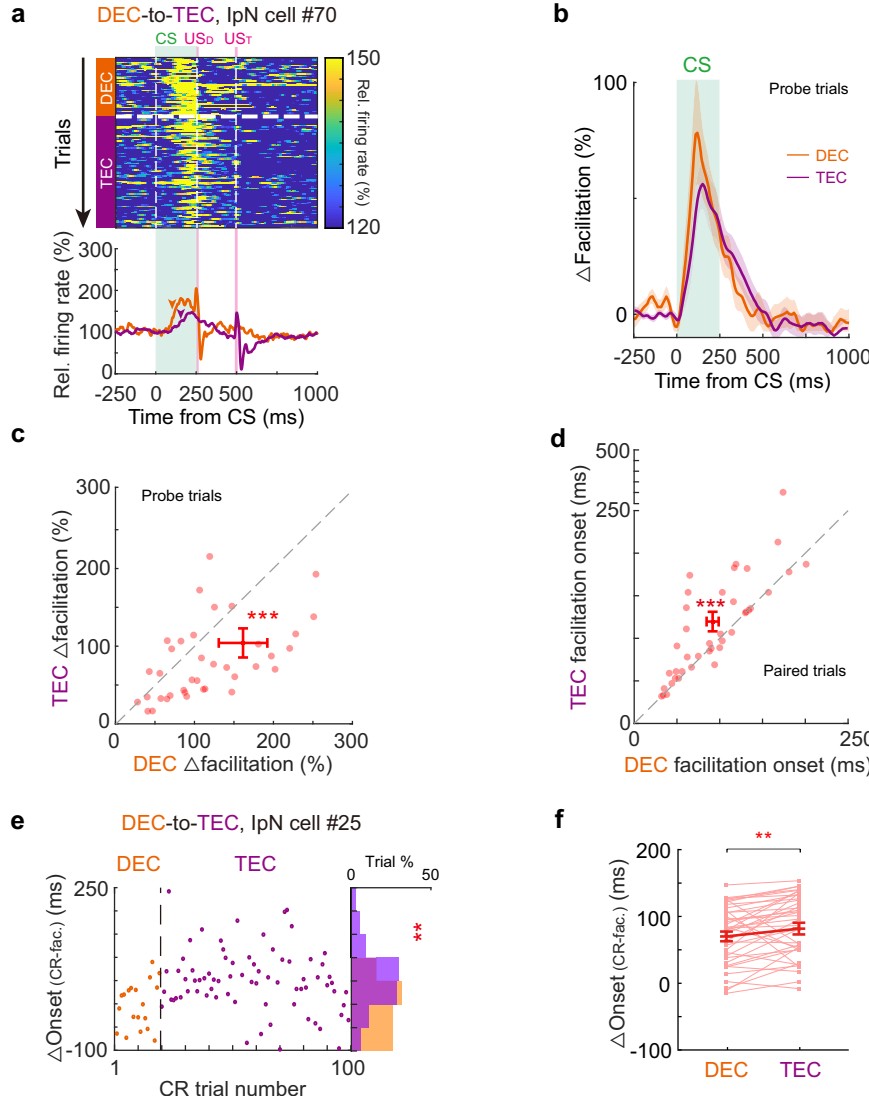

**Fig. 4 | Adaptation of CR timing is not fully explained by the changes of IpN modulation during DEC-to-TEC adaptation. a** Heatmap of trial-by-trial instantaneous firing rate (top) and the average firing rates (bottom) of an example IpN neuron during DEC-to-TEC adaptation. Arrowheads indicate the modulation onsets, $US_D$ and $US_T$ indicate the USs of DEC and TEC sessions. **b** Average normalized firing rates of all facilitated IpN cells ($n = 40$ cells from 27 sessions) during the recordings. **c** Pairwise comparison of facilitation amplitude during the DEC and TEC sessions for all IpN facilitation neurons ($P = 0.0003$, two-sided Wilcoxon test, $n = 40$ cells). **d** Same as (**c**), but for the facilitation onset timing ($P = 0.001$, two-sided

paired $t$-test, $n = 40$ cells). **e** Left, the trial-by-trial interval (Δ) between the onset timings of CR and neuronal modulation from an example IpN cell during DEC-to-TEC adaptation. The distribution histogram of the Δ value is shown on the right ($P = 0.0097$, two-sided Mann–Whitney test, $n = 19$ and 80 trials). **f** Summary of all facilitation IpN neurons indicating an increase in the average Δ value after DEC-to-TEC adaptation ($P = 0.0053$, two-sided Wilcoxon test, $n = 40$ cells). Data are shown as the mean ± s.e.m., **$P \le 0.01$, and ***$P \le 0.001$. Source data are provided as a Source data file.

activity of IpN neurons did not fully correlate with this behavioral adaptation (Figs. 3–6). Moreover, the temporal adaptation occurred within a mere few trials, significantly faster than the time needed for associative learning that hinges on cerebellar long-term plasticity[8,9,12]. We posited that other brain regions might play a role in the rapid adaptation of CR timing. Previous studies have underscored the mPFC's significance in TEC acquisition and expression[14,16,29–36]. We sought to investigate whether the mPFC could be instrumental in orchestrating the bidirectional adaptation between TEC and DEC paradigms. We recorded the mPFC neurons located contralateral to the trained left eyes in mice during either TEC or DEC (Fig. 7a). Our recordings showed that 20.1% (64 out of 318) and 27.6% (94 out of 341) of mPFC neurons exhibited CR-related modulation in TEC- and DEC-trained mice, respectively (Fig. 7b, c, Supplementary Fig. 11a, b). Among these neurons, a subset exhibited transient facilitation locked

to the CS onset (Fig. 7d, f), with transient facilitation onsets of $111.63 \pm 8.71$ ms for the TEC group and $100.87 \pm 4.67$ ms for the DEC group. Several mPFC neurons displayed sustained modulation, and intriguingly, some of them exclusively modulated during CR trials, but not during non-CR trials (Fig. 7e, g, Supplementary Fig. 11c, d). This distinct pattern suggests that mPFC activity is closely associated with conditioned eyelid closure during DEC and TEC.

## Rapid temporal adaptation of CR onset timing is encoded in mPFC neurons

We then investigated the involvement of mPFC neurons during the adaptation of the two paradigms, DEC-to-TEC and TEC-to-DEC, respectively (Fig. 8a, g). Consistent with the previous findings (Figs. 3f and 5f), both cohorts of mice had rapid adaptation when exposed to the novel CS-US interval (Fig. 8b, h, $P < 0.0001$ for both,

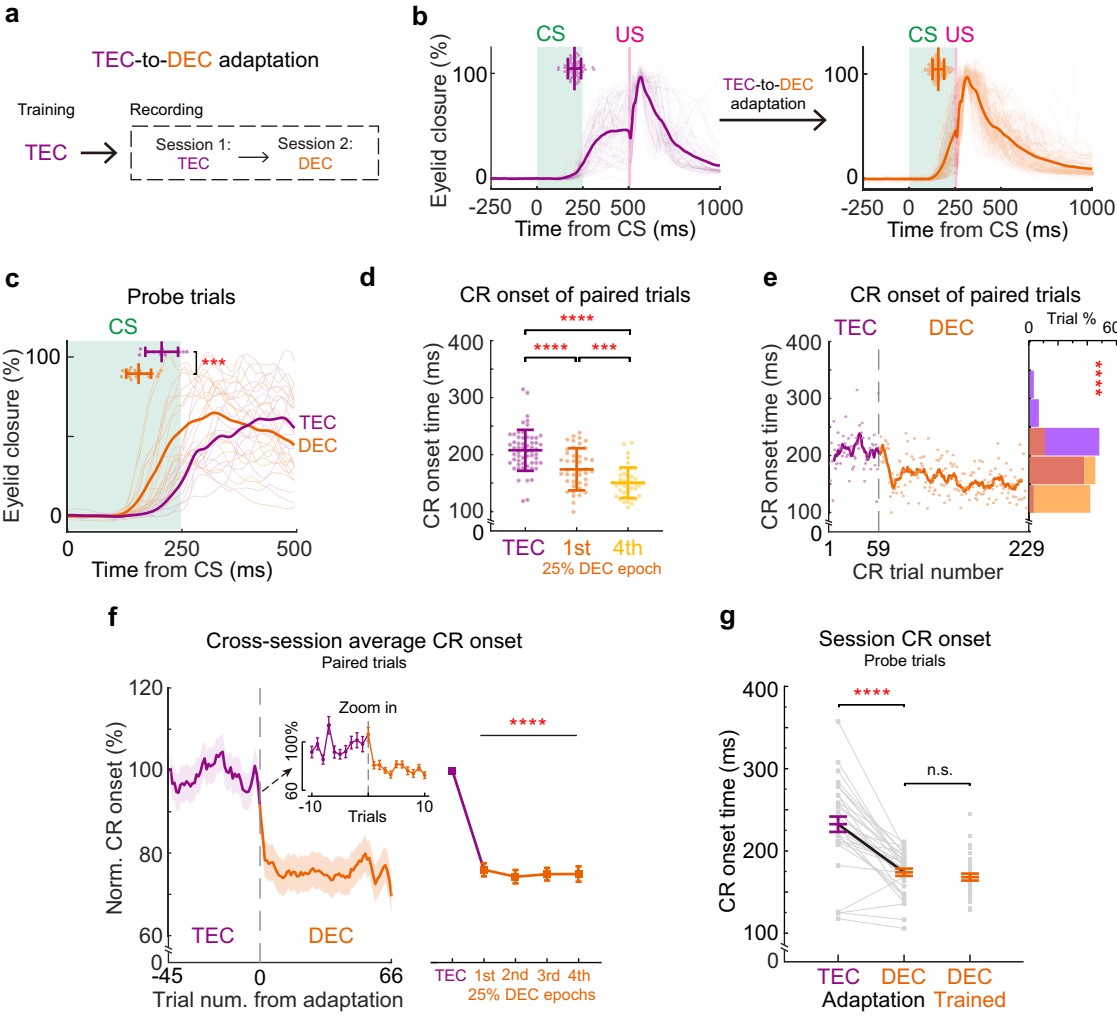

**Fig. 5 | Rapid adaptation of CR onset timing during the TEC-to-DEC adaptation paradigm. a** Experimental procedure for the animal training and paradigm switch from TEC to DEC. **b** Eyelid closure curves during the TEC (left, $n = 58$ trials) and DEC (right, $n = 171$ trials) sections of the TEC-to-DEC adaptation. The scatter plots at the top show the CR onsets of individual trials and the session average. **c** Average and individual (in dark and light colors) probe trial eyelid closure curves in the TEC (purple) and DEC (orange) sessions of the same example recording in (**b**). The scatter plots at the top show the CR onsets of each trial ($P = 0.0002$, two-sided Mann–Whitney test, $n = 11$ and 19 trials). **d** Paired trial CR onsets from the same example session, which is divided into TEC epoch (purple), the first 25% DEC epoch (orange), and the last 25% DEC epoch (light orange) (TEC vs. first and last 25% DEC $P = 2.3 \times 10^{-5}$ and $P = 7.1 \times 10^{-14}$, first vs. last 25% DEC $P = 0.001$, two-sided Mann–Whitney test, $n = 58$, 43, and 42 trials). **e** CR onsets of each trial during the example recording from TEC to DEC indicating rapid adjustment. The curve illustrates the

moving average. The distribution histogram of the CR onsets in TEC (purple) and DEC (orange) is shown on the right (CR onset time TEC vs. DEC $P = 1.5 \times 10^{-15}$, two-sided Mann–Whitney test, $n = 58$ and 171 trials). **f** Left, cross-session averages of normalized CR onset timing from all TEC-to-DEC adaptation experiments. The inset panel shows the zoom-in data before and after the paradigm switch. Right, statistical analysis of the CR onset timing for the TEC sessions and all 4 quartiles of DEC sessions ($P < 1.0 \times 10^{-15}$, two-sided repeat measurement one-way ANOVA, $n = 33$ sessions from 14 mice). **g** Comparisons of CR onset timing of all TEC-to-DEC adaptation experiments together with that of the DEC-trained mice ($P = 2.05 \times 10^{-8}$ and 0.14, two-sided Wilcoxon test and Mann–Whitney test, $n = 33$ and 38 sessions). Data are shown as the mean ± s.e.m., except for (**b**–**d**), which are the mean ± SD, n.s.: not significant, ***$P \le 0.001$, and ****$P \le 0.0001$. Source data are provided as a Source data file.

$n = 11$ DEC-to-TEC and $n = 8$ TEC-to-DEC sessions). Concurrent with these changes in CR onset, a subset of mPFC neurons displayed decreased modulation amplitudes during the DEC-to-TEC adaptation (Fig. 8c, d, $P = 0.0001$ for CR onset and $P = 1.19 \times 10^{-8}$ for modulation, $n = 65$ DEC, and $n = 71$ TEC trials in Fig. 8c for the example recording, $P = 0.0005$, $n = 29$ cells in Fig. 8d for population). Interestingly, another subpopulation of mPFC neurons showed a higher baseline firing in response to the adaptation (Fig. 8e, f, $P = 0.02$ for CR onset and $P = 1.32 \times 10^{-13}$ for baseline firing rate, $n = 36$ DEC, and $n = 82$ TEC trials in Fig. 8e for the example recording, $P < 1.0 \times 10^{-15}$, $n = 133$ cells in Fig. 8f for population). During TEC-to-DEC recordings, we also identified mPFC neurons displaying either decreased modulation or increased baseline firing rate (Fig. 8i–l) throughout the DEC epoch ($P < 1.0 \times 10^{-15}$ for CR onset and $P = 0.049$ for

modulation, $n = 41$ TEC, and $n = 38$ DEC trials in Fig. 8i for example recording, $P = 0.00039$, $n = 12$ cells in Fig. 8j for population; $P < 0.0001$ for both CR onset and baseline firing rate, $n = 41$ TEC, and $n = 38$ DEC trials in Fig. 8k for the example recording, $P < 0.0001$, $n = 75$ cells in Fig. 8l for population). In summary, our results illustrate altered CR-related modulations and baseline activities in mPFC neurons during paradigm adaptation, implying that mPFC neurons could potentially signal the paradigm switch and guide the adaptive changes in CR timing.

## The mPFC activity is essential for the temporal adaptation of CRs

To illustrate the functional relevance of the mPFC in instructing the temporal adaptation, we injected GABA$_A$ receptor agonist muscimol,

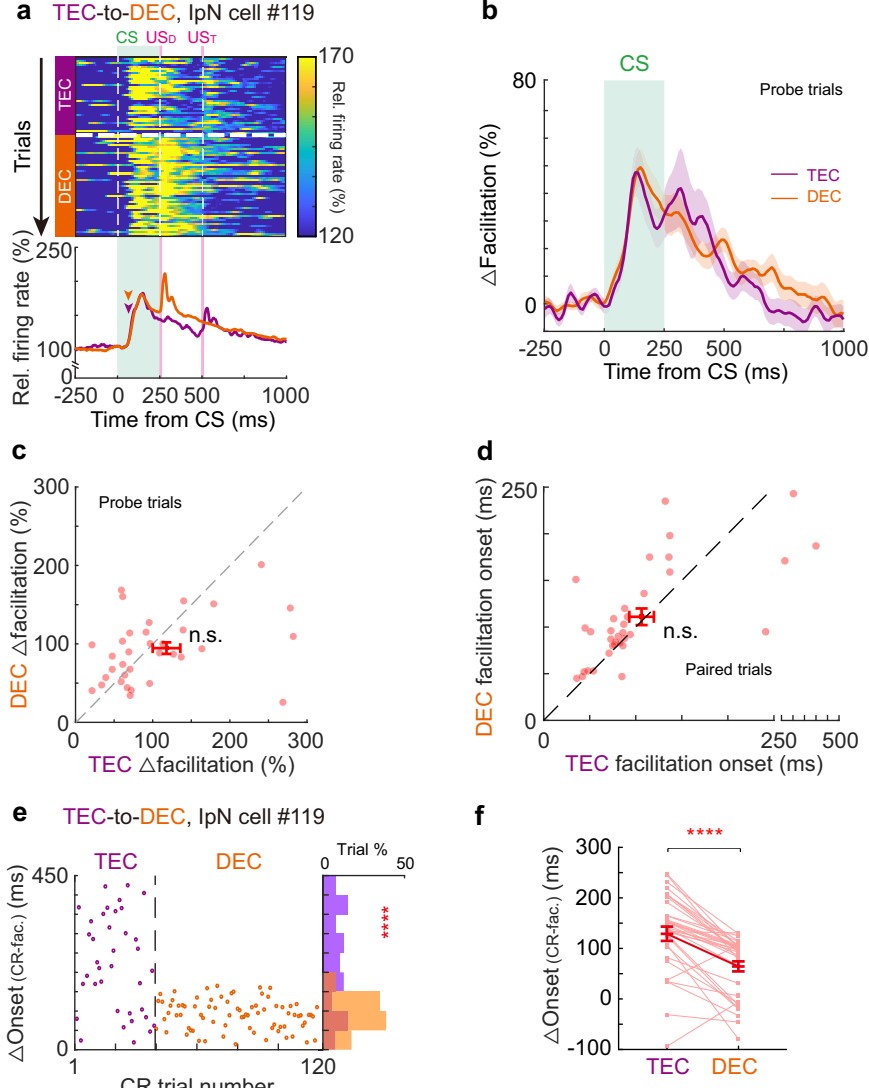

**Fig. 6 | Adaptation of CR timing is not explained by the change of IpN modulation during TEC-to-DEC adaptation. a** Heatmap of trial-by-trial instantaneous firing rate (top) and average firing rates (bottom) of an example IpN neuron during the adaptation from TEC (purple) to DEC (orange). Arrowheads indicate the onsets of spike modulation, $US_D$ and $US_T$ indicate the USs of DEC and TEC sessions. **b** Average normalized firing rates of all facilitated IpN cells ($n = 35$ cells from 25 sessions) during the TEC and DEC sessions. **c** Pairwise comparison of facilitation amplitude during the TEC and DEC sessions for all IpN facilitation neurons ($P = 0.40$, two-sided Wilcoxon test, $n = 35$ cells). **d** Similar to (**c**) but for the

facilitation onset timing ($P = 0.65$, two-sided Wilcoxon test, $n = 35$ cells). **e** Left, the trial-by-trial interval ($\Delta$) between the CR and neuronal modulation onset timings from an example IpN cell during TEC-to-DEC adaptation. The distribution histogram of the $\Delta$ value is shown on the right ($P = 5.24 \times 10^{-6}$, two-sided Mann–Whitney test). **f** Summary of all facilitation IpN neurons indicating a decrease in the average $\Delta$ value after TEC-to-DEC adaptation ($P = 1.07 \times 10^{-5}$, two-sided Wilcoxon test, $n = 35$ cells). Data are shown as the mean ± s.e.m., n.s.: not significant, ****$P \leq 0.0001$. Source data are provided as a Source data file.

in the mPFC of either DEC or TEC-trained mice before the paradigm switch (Fig. 9a, b, Supplementary Fig. 12a, b). With targeted bilateral muscimol injections in the mPFC, we observed a reduction in CR percentage, although not complete elimination, in both DEC- and TEC-trained mice (Supplementary Fig. 12c, d). Importantly, bilaterally inhibiting the mPFC was sufficient to abolish the rapid adaptation of CR onsets during both the DEC-to-TEC and TEC-to-DEC adaptations (Fig. 9c–f, in Fig. 9d, $P = 0.97$ and $0.054$, respectively, $n = 11$ sessions, in Fig. 9f, $P = 0.89$ and $0.45$, respectively, $n = 15$ sessions). These findings support the functional involvement of the mPFC in regulating the temporal adaptation of CRs, suggesting that the frontal cortex and cerebellum might work in tandem to dynamically adjust the precise relationship between sensory input and motor output, allowing for the flexible modulation of behavior based on specific task requirements (Fig. 9g).

## Discussion

The dynamic regulation of action timing plays a crucial role in guiding appropriate movements, yet the neural mechanisms regulating the temporal feature of sensorimotor behaviors remain largely elusive. In this study, we leveraged the well-understood neuron circuits underlying the sensory-cued eyelid closure response, and examined the neuronal characteristics of cerebellar and mPFC circuits for the precise temporal regulation of conditioned responses. Focusing on animals trained with identical CS duration but distinct CS-US intervals, our study reveals the unique cerebellar coding features for the DEC and TEC paradigms (Figs. 1 and 2). By examining the neural dynamics during DEC-to-TEC and TEC-to-DEC adaptations, we uncovered the characteristics of cerebellar activity underlying the rapid adaptation of action timing (Figs. 3–6). We further established the essential role of the mPFC in mediating such bidirectional adaptation (Figs. 7–9). All

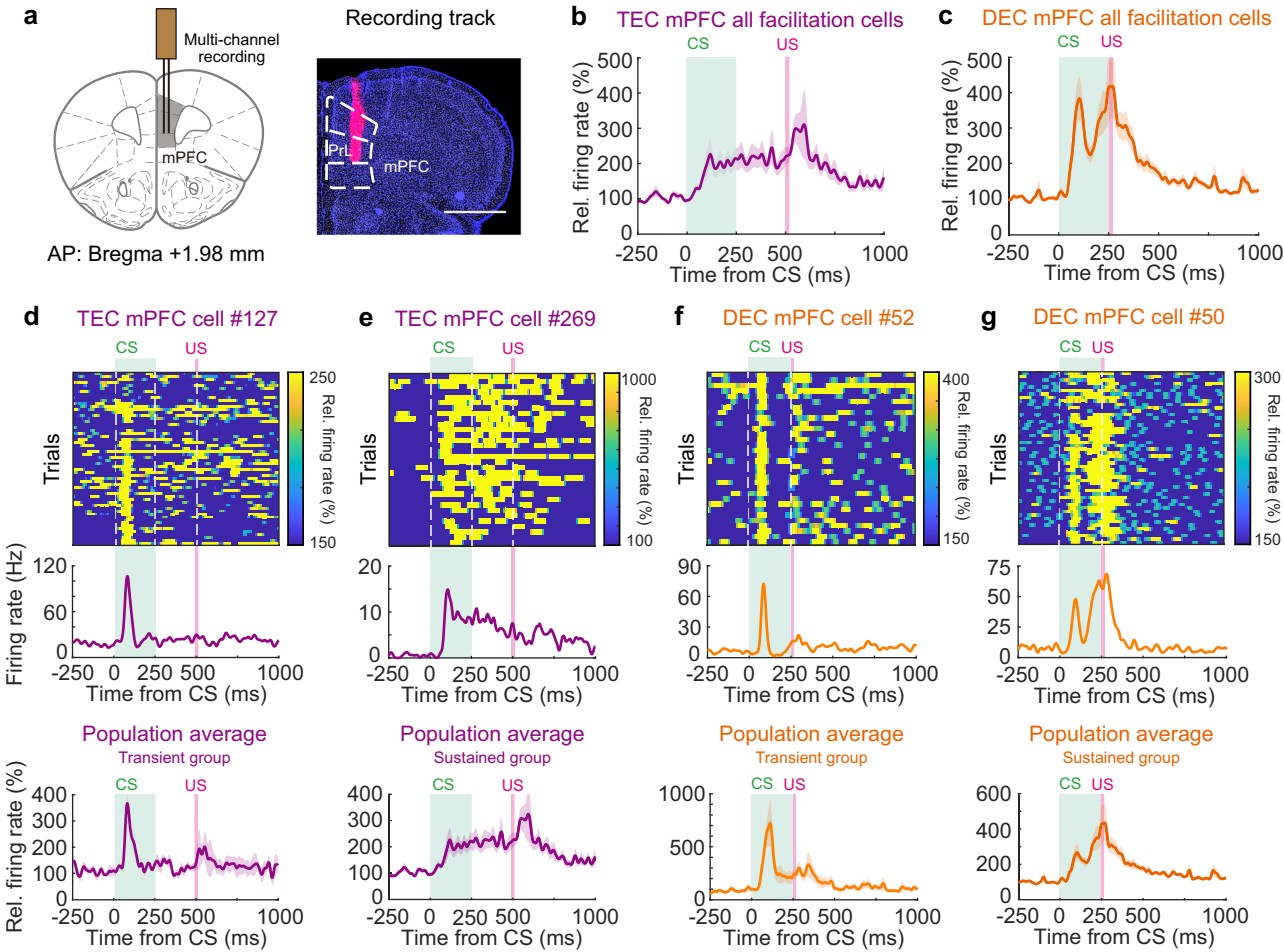

**Fig. 7 | TEC- and DEC-related modulation observed in the mPFC. a** Left, schematic of the mPFC recording site projected to the mouse brain atlas. Right, an example DiI-labeled recording track (PrL, prelimbic area, scale bar, 1 mm). **b** Average relative firing rates of all mPFC cells showing CR-related activities during TEC recordings ($n = 64$ cells, from 3 mice). **c** Similar to (**b**) but for the DEC recordings ($n = 94$ cells, from 3 mice). **d**–**g** Top and middle, the instantaneous firing rate heatmaps (top) and session average firing rate (middle) of example mPFC neurons, two cells during TEC and two cells during DEC. Bottom, population

average of mPFC neuronal activity with specific modulation patterns during TEC and DEC recordings. **d**, **e** Neurons with transient (**d**) or sustained (**e**) facilitation recorded during the TEC paradigm ($n = 5$ and 59 cells). **f**, **g** Neurons with transient (**f**) or sustained (**g**) facilitation during the DEC paradigm ($n = 18$ and 78 cells). Data are shown as the mean ± s.e.m. Panel **a** adapted from this article was published in The Mouse Brain in Stereotaxic Coordinates (2nd edition), Keith B. J. Franklin and George Paxinos, Page 64, Copyright Elsevier Academic Press (2001)[101].

these findings support the role of cerebral and cerebellar circuits for precise and adaptive temporal control of sensorimotor behaviors.

## Multimodal IpN coding for DEC and TEC paradigms
We have extensively examined the neuronal and behavioral dynamics in mice trained using either the DEC or TEC paradigms. The fundamental objective of Pavlovian eyeblink conditioning is to attain precise eyelid closure timing before the presentation of the US. A wealth of studies has illuminated that this anticipatory response is facilitated by task-related modulation in IpN neurons, which occurs during the learning process[8–11,13–16,18–22,24–26,42–46]. Given that IpN neurons directly influence downstream premotor neurons in the red nucleus, it is postulated that CR-related modulation in IpN neurons is closely correlated with CR timing[9,11,24,26]. Our comprehensive trial-by-trial and decoding analyses investigating the relationship between CR onset timing and IpN facilitation unequivocally validate the predictive role of IpN neurons. Notably, the IpN facilitation predicts the timing and kinematics of CR (Figs. 1 and 2). Therefore, our findings support the perspective that IpN neurons play a pivotal role in the regulation of CR timing and kinematics in both the DEC and TEC paradigms.

Whether IpN employs a common coding strategy to regulate CR timings in both DEC and TEC paradigms has not been systematically explored in mice. Mice trained in the DEC and TEC paradigms, which comprise an identical CS but different CS-US intervals, differ in their CR timings but not CR amplitudes (Fig. 1). As CR-related modulation in the IpN is considered the key driver of eyelid closure[9,11], one would assume that the IpN modulation in DEC- and TEC-trained mice also differs in timing but not in amplitude. A detailed activity-behavior correlation analysis revealed, however, that DEC-trained mice had larger task-related modulation amplitudes. In contrast, the facilitation onset timings of IpN neurons were comparable between TEC- and DEC-trained mice (Fig. 2), resulting in a much longer delay between IpN modulation and CR onset in TEC mice. Focusing on the 'eyeblink neurons', which were defined by their significant trial-by-trial activity-behavior correlation, we still observed a clear difference in the activity-behavior intervals between TEC- and DEC-trained mice (Supplementary Fig. 2). The delays between neural activity and behavioral response were an order of magnitude longer than the synaptic delays observed in CR onset following direct electrical stimulation of the IpN (Supplementary Fig. 3). These data suggest that information conveyed by the

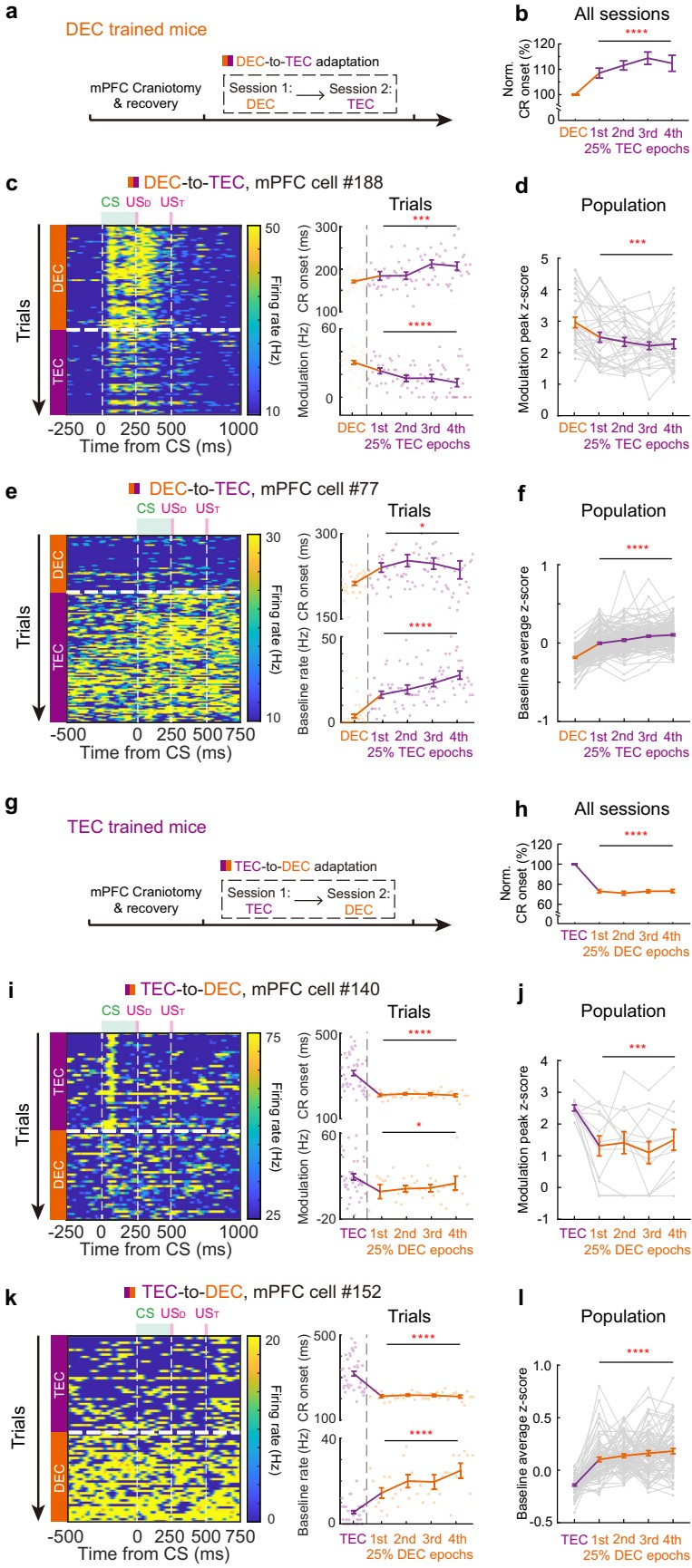

**Fig. 8 | Adaptation of the CR-related modulations and baseline activities of mPFC neurons during paradigm switching. a** Experimental procedures of DEC-to-TEC adaptation with mPFC recordings. **b** Statistical analysis of the CR onset timing for the DEC epochs and all four quartiles of TEC adaptation epochs with mPFC recording cohort ($P = 9.84 \times 10^{-5}$, two-sided repeat measurement one-way ANOVA, $n = 11$ sessions). **c** Left, firing activity heatmap of an example neuron with significantly reduced CR-related modulation during the TEC epoch of DEC-to-TEC adaptation. Right, the trial-by-trial CR onset (top, $P = 0.0001$) and the modulation (bottom, $P = 1.19 \times 10^{-8}$, two-sided Brown-Forsythe ANOVA test) during the same recording session ($n = 65$ DEC, and $n = 71$ TEC trials). The changing trend is shown by the mean values of the DEC epoch and four quartiles in the TEC epoch. **d** The z-scored modulation peak during the DEC epoch and four quartiles in the TEC epoch with the mPFC neurons showing decreased modulation in TEC epochs of the adaptation ($P = 0.0005$, two-sided repeat measurement one-way ANOVA, $n = 29$ cells). **e, f** Similar to (**c, d**), but for neurons with increased baseline firing rate during the TEC epoch of DEC-to-TEC adaptation ($P = 0.023$ for CR onset and $P = 1.32 \times 10^{-13}$

for baseline firing rate, two-sided Brown-Forsythe ANOVA test, $n = 36$ DEC, and $n = 82$ TEC trials in (**e**), $P < 1.0 \times 10^{-15}$, repeat measurement one-way ANOVA, $n = 133$ cells in (**f**)). **g** Experimental procedures of TEC-to-DEC adaptation with mPFC recordings. **h** Similar with (**b**) but for TEC-to-DEC adaptation with mPFC recording cohort ($P = 1.9 \times 10^{-14}$, two-sided repeat measurement one-way ANOVA, $n = 8$ sessions). **i, j** Similar to (**c, d**), but for reduced modulation in the DEC epochs of TEC-to-DEC adaptation ($P < 1.0 \times 10^{-15}$ for CR onset and $P = 0.049$ for modulation, two-sided Brown-Forsythe ANOVA test, $n = 41$ TEC, and $n = 38$ DEC trials in (**i**), $P = 0.0004$, two-sided repeat measurement one-way ANOVA, $n = 12$ cells in (**j**)). **k, l** Similar to (**e, f**), but for neurons with increased baseline firing rate during the DEC epoch of TEC-to-DEC adaptation ($P < 1.0 \times 10^{-15}$ and $P = 4.35 \times 10^{-7}$ for CR onset and baseline firing rate, two-sided Brown-Forsythe ANOVA test, $n = 41$ TEC, and $n = 38$ DEC trials in (**k**), $P < 1.0 \times 10^{-15}$, two-sided repeat measurement one-way ANOVA, $n = 75$ cells in (**l**)). Data are shown as the mean ± s.e.m. *$P \le 0.05$, ***$P \le 0.001$, and ****$P \le 0.0001$. Source data are provided as a Source data file.

IpN is complex and cannot be viewed as merely a 'motor command for eyelid closure' and instead reflects a multimodal encoding pattern.

The complexity of these signals could stem from both cerebellar and extracerebellar processing[47,48]. During EBC, CS- and US-related inputs converge at the cerebellar cortex, inducing persistent, input-specific synaptic depression at the parallel fiber to PC synapses[49]. The resultant suppression of simple spike activity in PCs, driven by long-term depression, is considered a central driver of the CRs[6,46]. Previous studies have suggested that the CR-related activities of PCs in animals transitioning from DEC to TEC paradigms share remarkable similarities and can be explained by a common inverse firing rate model[6]. However, more recent research challenges this notion, suggesting that both facilitatory and suppressive modulations occur in PCs during DEC[50]. Consistently, we observed both task-related facilitation and suppression in the IpN neurons of both DEC- and TEC-trained mice. This challenges the current view that IpN modulation directly encodes motor command, instead favoring the possibility of multiplex coding for precise and adaptive control. Beyond the sensorimotor regions, extracerebellar regions such as the hippocampus, mPFC, and striatum have been implicated in contributing to TEC expression[8,37,51–54]. The mossy fiber-granule cell pathway, which conveys these cognitive signals, could provide additional task-related information to the IpN. A third stream of task-related information might stem from nucleo-cortical feedback loops, which have been proposed to offer predictive signals during DEC training[55–57]. These diverse input patterns might be beneficial for extending high-dimensional representations at the cerebellar input layer, thereby supporting behavioral state-specific computation in IpN[58]. In addition, converging mossy fiber and climbing fiber collaterals onto the IpN could carry supplementary CS- and US-related information[11,59]. This is substantiated by studies showing that mossy fiber collaterals undergo structural[59] and synaptic plasticity[60], thus directly shaping IpN neuron activities during behavior. In summary, these circuit mechanisms collectively control the encoding dynamics in cerebellar output, leading to a richer and more nuanced signaling landscape.

### Cortico-cerebellar communication and rapid temporal adaptation

CR timing is determined by the specific temporal relationship between the CS and US during learning[61,62]. Recent research indicates that cerebellar granule cells, optimized for encoding high-dimensional representations, form the foundation for temporal information encoding in sensorimotor behaviors[58]. Learning-triggered long-term plasticity at the synapses between parallel fibers and PCs is responsible for selecting relevant inputs, thereby shaping PC output patterns[46]. It might be expected that, in response to an abrupt change of CS-US interval, animals should temporarily maintain their CR onset timing, and gradually adjust to the new timing over training, following the time

course of long-term synaptic plasticity. Surprisingly, our findings counter this expectation, as mice trained under DEC or TEC paradigms exhibited the capacity to instantly adapt CR timings within a single training session upon the introduction of novel CS-US intervals (Figs. 3 and 5). Such swift adaptation can manifest within just a few trials, significantly faster than the temporal dynamics of long-term synaptic plasticity in the cerebellum[63]. During this rapid adaptation, other cortico-cerebellar mechanisms are likely at play. In the conventional framework, CR timing in DEC primarily arises from cerebellar computation, while TEC engages an additional mPFC-cerebellar pathway that bridges the CS-free interval[33,37]. The sequential activation of two sets of mossy fibers, encoding sensory and mPFC inputs, collectively shapes the CR's temporal profile during TEC[62]. The latter set may be selectively disengaged, whereas the cerebellar computation of the sensory pathway prevails during TEC-to-DEC adaptation. To this end, cerebellar computation is expected to be similar in TEC-to-DEC converted mice and DEC-trained mice. This is supported by the results that the CR timing after TEC-to-DEC adaptation was identical to the CR timing of the DEC mice (Fig. 5g). Following the same reasoning, it is possible that novel extracerebellar information, which emerges during DEC-to-TEC adaptation, drives additional adaptation of IpN activity (Fig. 4) and delays the CR onset timing (Fig. 3). However, the DEC-to-TEC adaptation is unlikely to be fully dependent on the mPFC-cerebellar pathway. Mice trained with DEC were never exposed to the stimulus-free interval before adaptation; therefore, it is highly unlikely that they would activate the appropriate set of mossy fibers to bridge the CS-free interval[33,37]. Yet, the switch in CR onset timing was as rapid as that seen in TEC-to-DEC adaptation. This finding contrasts sharply with the previous DEC-to-TEC studies in rabbits, where rabbits required prolonged training to gradually extinguish the original DEC response and reacquire a TEC response[6]. In line with our data using different TEC intervals (Supplementary Fig. 7), we conclude that rodents possess the ability to adapt the timing of acquired CR responses within a single training session without the need for the extensive process of extinction of old CR timing and reacquisition of new CS-US intervals.

We show that the mPFC was critically involved in modulating CR expression as well as in facilitating rapid DEC-to-TEC and TEC-to-DEC adaptations (Figs. 7–9), consistent with the prominent roles of mPFC in TEC learning[14,16,29–36]. This study sheds light on novel roles of the mPFC in both TEC and DEC paradigms. The phasic and persistent modulation patterns in the mPFC align with previous findings from TEC-trained rabbits[33,36], suggesting a multiplex coding of sensorimotor association in this region. This complexity arises from the coexistence of multiple components, including sensory, motor command, and outcome information[33,36,64]. Interestingly, prominent mPFC modulation time-locked to CS onset was observed not only in TEC-trained mice but also in DEC-trained mice. Furthermore, when we pharmacologically

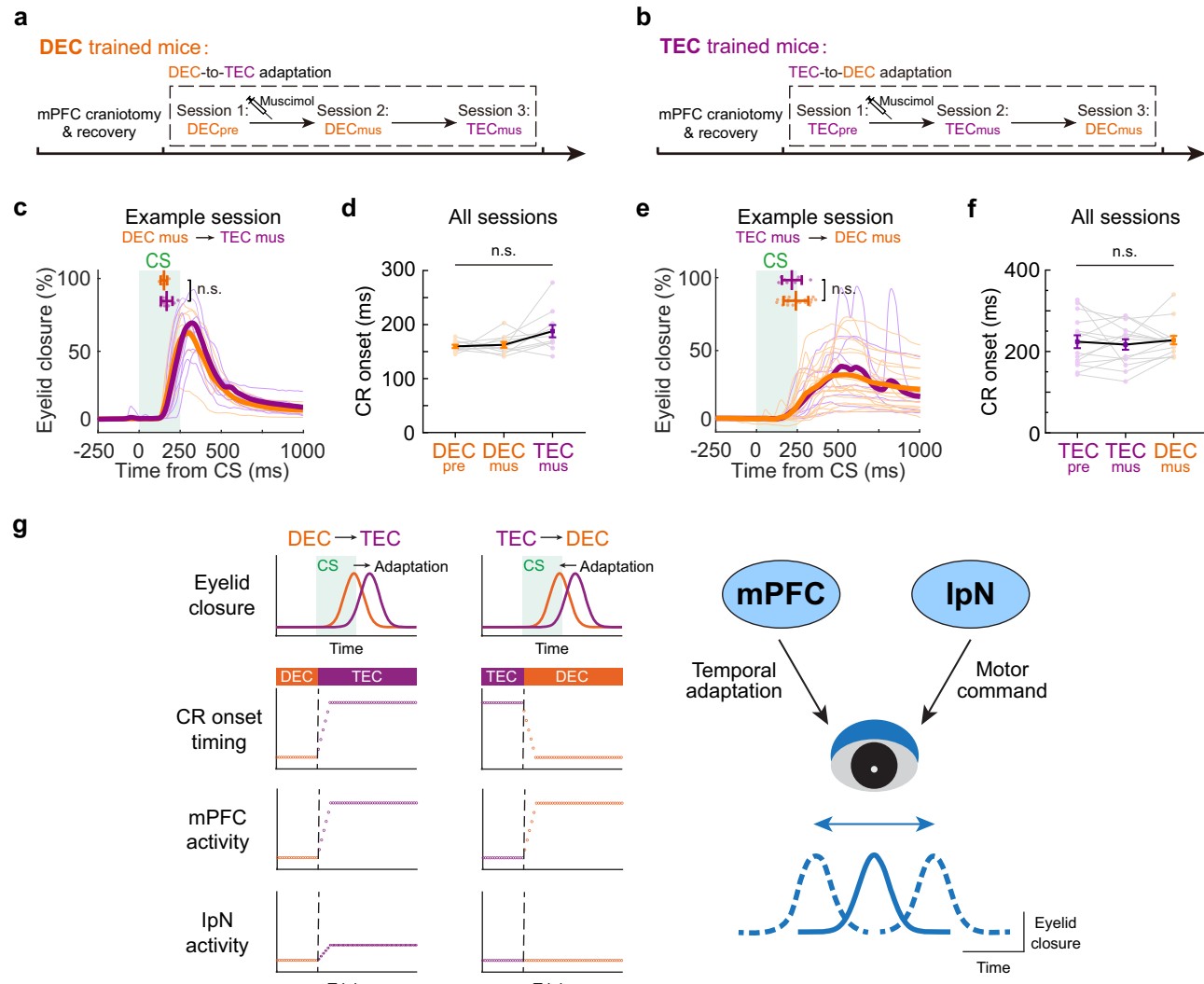

**Fig. 9 | mPFC activity is essential for the adaptation of CR timing during paradigm switching. a** Experimental procedures of mPFC inhibition during DEC-to-TEC adaptation. The recordings include three epochs: DEC before muscimol injection (DECpre), DEC after muscimol injection (DECmus), and the paradigm switch to TEC (TECmus). **b** Similar to (**a**) but for the TEC-to-DEC adaptation. **c** Eyelid closure curves of the CS-only trials during DEC (orange) and TEC (purple) sessions from an example session. The scatter plots at the top show the CR onsets of individual trials ($P = 0.47$, two-sided Mann–Whitney test, $n = 7$ and 9 trials). **d** Summary of CR onset times for all mPFC inhibition during DEC-to-TEC sessions ($P = 0.97$ and 0.054, two-sided Wilcoxon test, $n = 11$ sessions from 3 mice). **e** Same as (**c**), but for an example TEC-to-DEC session ($P = 0.44$, two-sided Mann–Whitney test, $n = 9$ and 20 trials). **f** Same as (**d**) but for all TEC-to-DEC sessions ($P = 0.89$ and 0.45, two-sided Wilcoxon test, $n = 15$ sessions from 3 mice). **g** Summary schematic of mPFC and IpN activity changes during two adaptation paradigms (left) and the putative functions of these two regions in adaptation (right). The CR timing could be rapidly adjusted in both directions in mice, which coincides with significant adaptation of neurons in the mPFC region, but less so in the cerebellar IpN region. We propose that mPFC and cerebellum serve distinct functions, which synergistically mediate the flexible adaptation of motor timing. Data are shown as the mean ± s.e.m., except for (**c, e**), which is the mean ± SD, n.s.: not significant. Source data are provided as a Source data file.

inhibited the mPFC, a suppression of CR percentage was observed in both DEC- and TEC-trained mice (Supplementary Fig. 12). These results do not necessarily contradict the notion that the mPFC is involved in TEC but not DEC learning[29,45,65–67]. Although mPFC activity may not be necessary for DEC acquisition because the CS is directly contingent on the US, it could still influence animals' internal states[68–70], which has been shown to contribute to CR expression in DEC-trained animals[71,72].

Our findings also offer new insights into the mechanisms of temporal adaptation of the mPFC. The mPFC is known to play a significant role in various high-order cognitive functions, including working memory[73–77], attention[78,79], and sensorimotor integration[80,81]. Given the complexity of mPFC activity during cognitive control tasks, it is oversimplified to assume that the sole function of mPFC in eye-blink conditioning is to bridge the CS-US interval by sending temporal information to the cerebellum via the pontine mossy fiber system.

Neither mPFC nor cerebellar nuclei neurons were found to fully adapt the modulation patterns following the changes of CR timing after the adaptation paradigms. It is therefore possible that in addition to the pathway from the mPFC to the cerebellum via the pontine nuclei, mPFC neurons that project various cortical and subcortical areas[82–85] may also play substantial roles in controlling the execution of adaptive movements. Two particularly noteworthy structures involved in this process are the thalamo-cortical and cortico-basal ganglia pathways[86]. The ability of the thalamo-cortical circuit to modulate rule switching has been observed in other tasks that demand flexible adjustments in both cue-specific and task-specific activities[28,87–90]. Moreover, the mPFC might regulate CR onset timing via the basal ganglia by balancing motor activation and inhibition[91–93]. The cortico-basal ganglia pathway is crucial for sensorimotor integration and adaptation of well-learned behaviors, especially for selectively promoting or postponing

the action timing[94–96]. During DEC-to-TEC adaptation, animals are required to withhold action, while during TEC-to-DEC adaptation, they need to actively reduce the action delay. The dynamic equilibrium between activation and inhibition of action planning, orchestrated by different mPFC-basal ganglia circuits, may provide flexibility for controlling action timing. Both the thalamo-cortical and cortico-basal ganglia pathways can functionally interact with the red nucleus in movement control[97–100], which parallel the cerebellar to red nucleus pathway. We hypothesize that the red nucleus may integrate the outputs from the cerebellar, cerebral, and basal ganglia to command the downstream facial nucleus motor neurons to execute the well-timed eyelid closure. Future research should investigate the communication across multiple brain regions and their collective roles in regulating adaptable sensorimotor behaviors.

## Methods

### Animals
Wild type C57Bl/6 mice aged 12 to 20 weeks were included in the experiments (both male and female) and housed individually (12:12 light/dark cycle, room temperature, and 40–70% humidity) with food and water ad libitum. The experiments were approved by the institutional animal welfare committee of Erasmus MC. In total, 24 mice were trained with the TEC paradigm, and 20 mice were trained with the DEC paradigm for electrophysiological recordings and behavior manipulation. We also included a dataset of IpN recordings in seven DEC-trained mice from our previous study[11].

### Surgery
Mice were anesthetized with 5% isoflurane for induction, and 2% isoflurane for maintenance, and kept at 37 °C using a heating pad during surgery. After fixing the animal into a standard mouse stereotaxic surgical plate (Stoelting Co., Wood Dale, IL, USA), we exposed the skull and applied Optibond All-In-One (Kerr, Scafati SA, Italy) for better pedestal attachment. A small brass pedestal was attached to the skull and fixed with Charisma (Heraeus Kulzer GmbH, Hanau, Germany). We performed craniotomy (-1.5 mm in diameter) above the target region (IpN or mPFC) for electrophysiology, electrical stimulation, and muscimol injection. The stereotaxic coordinates for the craniotomy over the IpN are AP: −2.2 mm (measured from Lambda), ML: 1.8 mm; for the craniotomy over the mPFC: AP: +1.9 mm (measured from Bregma), ML: 0.4 mm. A small chamber was made by attaching Charisma around the skull of the craniotomy and finally closed by a low-viscosity elastomer sealant (Kwik-cast, World Precision Instruments, Sarasota, FL, USA). Bupivacaine hydrochloride (2.5 mg mL⁻¹, i.p.) was injected after surgery.

### Behavioral training
After a one-week recovery from the surgery, mice were head-fixed on a cylindrical treadmill in a sound- and light-attenuated setup for one week of habituation[7]. For DEC and TEC training, a green LED (CS) was placed ~7 cm in front of the mouse, while a corneal air puff (US, tip opening Φ = 0.8 mm, 30 psi) was directed at the left eye with 5 mm distance. The CS duration was 250 ms in both TEC and DEC paradigms. A 15-ms US aair puff was delivered at the CS offset for the DEC paradigm but with a 250 ms interval after the CS offset for the TEC paradigm. Randomized inter-trial intervals of 8–12 s were employed, with a minimum of 200 trials per training session. We introduced CS-only trials randomly within the normal paired trials at a 1:6 ratio. Animals achieving a high CR-trial probability (70% of total trials) were considered as well-trained. Eyelid movement was captured by a 250-fps camera (scA640-120gm, Basler, Ahrensburg, Germany). National Instruments System (NI 9263 and NI 9269, National Instruments, Austin, TX, USA) was used to control the triggers and digitization of camera signals using custom LabView codes. The RHD2000 Evaluation System (Intan Technology) with a 20 kHz sample rate recorded digitized triggers and eyelid positions.

### In vivo electrophysiology
After a recovery period of 2–3 days from the craniotomy, we conducted in vivo electrophysiological recordings on awake-behaving mice. We performed acute recordings throughout the study using acute 64-channel silicon probes (ASSY 77H2, Cambridge NeuroTech). We vertically inserted the probe into either the IpN (at approximately 2.4 mm depth) or mPFC (at -1.5 mm depth), guided by an electrode manipulator (Luigs and Neumann SM7, Germany). Neuronal signals were notch-filtered at 50 Hz, amplified, and digitized by an Intan RHD2000 Evaluation System (Intan Technology) at a 20 kHz sampling rate and were further analyzed offline using custom-written Matlab codes. For the electrophysiological experiments during TEC-to-DEC and DEC-to-TEC adaptation, we recorded the activities of the same neurons for at least 50 trials before and after the paradigm switch. In general, we recorded for 5 days for each recording region for all types of behavior paradigms, except for the adaptation paradigms, which were recorded for 2 days to avoid long-term learning. On the last day of recording, a fluorescent DiI-coated probe was penetrated to label the recording track followed by histology and imaging.

### IpN electrostimulation
Data was acquired according to our previous study[12]. Briefly, after IpN craniotomy, a stimulation glass electrode (8 µm tip opening) with 2 M saline-0.5% alcian blue was penetrated into IpN. Electrical stimuli (500 Hz with 250 µs biphasic pulses and 200 ms pulse train) were performed to the animals with current strengths of 0.6, 0.8, 1.0, and 1.2 µA. Eyelid movement ipsilateral to the stimulation side was recorded using the devices mentioned above. The maximum eyelid closure during 1.2 µA stimulation was viewed as full eyelid closure.

### Pharmacological inhibition of mPFC
For mPFC inhibition during TEC-to-DEC adaptation, a small craniotomy was made over the bilateral mPFC in TEC-trained mice. We first recorded 50 TEC trials as a control, then gently injected 20 nL of muscimol (0.5 mg/mL, Tocris Bioscience, 0289) in the mPFC. Five minutes later, we switched to the DEC paradigm to test their CR adaptation. We performed a similar procedure for the DEC-to-TEC adaptation experiment.

### Histology and imaging
Animals were deeply anesthetized by injecting a pentobarbital sodium solution (50 mg/kg, i.p.) and transcardially perfused with saline and a solution of 4% paraformaldehyde (PFA) in 0.1 M phosphate buffer (PB, pH = 0.74). The brains were removed immediately and post-fixed in 4% PFA for 2 h, and then dehydrated in 10% sucrose in a 4 °C fridge overnight. On the next day, the brains were embedded in 12% gelatin with 10% sucrose, fixed again in a 10% formalin-30% sucrose solution for 2 h, and then dehydrated in 30% sucrose at 4 °C overnight. Coronal brain slices with a thickness of 50 µm were obtained using a microtome (SM2000R, Leica), and collected in 0.1 M PB. Brain slices with DiI labeling were stained with DAPI as background and imaged by Zeiss LSM700. Slices with alcian blue labeling were stained with neutral red as the background and scanned by NanoZoomer (Hamamatsu).

### Eyeblink data analysis
As described in previous work[11,12], the investigation of eyelid position changes in response to the CS and US involved normalizing each trial to a 500 ms baseline preceding the CS onset. We removed trials with a noisy baseline (spontaneous blinking) by performing an iterative Grubbs' outlier detection test ($\alpha = 0.05$) on the standard deviations of baseline. A CR trial was determined if the eyelid closure exceeded 5% of the mean baseline, and CR onset was defined as the timing at which eyelid closure exceeded six standard deviations of the baseline value. We also explored alternative criteria for detecting CR onset, based on CR amplitude[11] and CR velocity[39] from prior studies. Peak amplitudes

of CR and UR were detected as the maximum during the CS-US interval and a 100 ms window after the US, respectively. CR slope was defined as the angle of a linear fit between 10% and 90% of CR peak amplitudes. Maximum CR velocity was calculated with 10 ms bins.

### Electrophysiology analysis

We analyzed the electrophysiological recordings using custom Matlab code. Raw recordings were band-pass filtered between 300–3000 Hz to subtract noise and field potential signals. We extracted spike events with amplitudes that exceed three SDs of the baseline value. Multi-channel recordings were sorted using JRCLUST[89,90], and all spike times were stored for further analysis. We transformed each spike into a Gaussian kernel (21 ms kernel for IpN single units, and 41 ms kernel for mPFC single units) and aligned them with CS onset for the peristimulus time histograms (PSTH) calculation. Single-trial instantaneous firing rate was further smoothed by another 10 ms sliding window for trial-by-trial analysis. To ensure data reliability, only cells with over 20 CR trials were included in the dataset for analysis of cell modulation in response to the CS. For IpN neurons, CR-related modulation was detected from 10 ms after CS onset to 10 ms before US onset, considering potential epoch crossings in PSTH construction. The baseline firing rate was calculated as the mean frequency in a 500 ms window before the CS onset. Neurons with PSTH changes exceeding three SDs of the baseline frequency were considered as CR-related modulation neurons, and the first time point reaching the 3-SD threshold was the modulation onset. For mPFC neurons, we determined CR modulation by assessing the significance of the baseline firing rate to the firing rate between CS and US. Transient mPFC facilitation neurons were characterized as cells with a half-wave width constituting less than 20% of the CS-US interval, along with a modulation peak occurring before 200 ms. The remaining modulated mPFC neurons were classified as part of the sustained facilitation group.

### Decoding classifier analysis

The decoding classifier analysis was adapted from the neural decoding toolbox, as described previously[41]. We utilized the max correlation coefficient classifier to decode potential differences in firing patterns of IpN single units between early and late CR onset trials in TEC recordings. To execute this, we organized all trials from a single recording session based on their CR onset times, evenly splitting them into early- and late-onset subgroups. Within a time window ranging from −500 to 500 ms, we established bins of 100 ms with a step of 50 ms to cluster spikes. In each decoding cycle, 90% of the trials were randomly selected from both early and late groups as training datasets, and the rest 10% of both served as testing datasets. This process was repeated 20 times as resampling. The accuracy of decoding was calculated by determining the percentage of accurately distinguishing early and late trials within the testing dataset. To assess the statistical significance of the decoding outcome, this procedure was repeated 10 times using shuffled data. IpN neurons that exhibited a decoding difference of at least 100 ms in modulation peak times, while maintaining less than a 20% difference in modulation amplitude, were categorized as temporal modulation neurons (Supplementary Fig. 4b). Neurons with amplitude differences exceeding 20% were classified as amplitude modulation neurons (Supplementary Fig. 4c, d).

### Statistics

All the analysis and statistics were performed using MATLAB and GraphPad Prism. The average eyelid closure curve and within-session CR onset scatter were plotted as mean ± standard deviation (SD); other data were presented as mean ± standard error of the mean (s.e.m.). Wilcoxon signed-rank test was used to detect the group differences of paired samples, except for the IpN modulation onset comparison in

DEC-to-TEC and TEC-to-DEC adaptation experiments, where a paired *t*-test was applied. Mann–Whitney U test was performed to compare the group differences of independent samples. Repeated measures one-way ANOVA was used for analyzing the data continuously collected from the same subject. Pearson's r was used to detect the correlation between two continuously distributed variables (95% confidence interval and 80% power were used which led to an effect size of 8). All the statistical tests were two-tail based. $P \leq 0.05$ was considered as significantly different, and were annotated as *$P \leq 0.05$, **$P \leq 0.01$, ***$P \leq 0.001$, and ****$P \leq 0.0001$.

### Reporting summary

Further information on research design is available in the Nature Portfolio Reporting Summary linked to this article.

## Data availability

The raw data in this study are available from the corresponding author (Z. Gao) upon request, due to the size of dataset. Source data are provided with this paper.

## Code availability

The data acquisition codes created in Labview are available from the corresponding author (Z. Gao) upon request. The custom codes in Matlab for analysis are available via the link: https://github.com/ZhongRenNeuroscience/Eyeblink-TEC-DEC.

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

## Acknowledgements

The authors thank M. Rutteman (Department of Neuroscience, Erasmus MC) for managing the animals, S. Dijkhuizen (Department of Neuroscience, Erasmus MC) for providing the eyeblink conditioning training boxes, S. Yu (Department of Neuroscience, Erasmus MC) for technical support of the electrophysiology setup and MATLAB programming, H. Hasanbegovic, and C. Schafer (Department of Neuroscience, Erasmus MC) for the constructive inputs to this study. This work is supported by CSC fellowship (Z.R., CSC201907720092), Medical Neuro-Delta (C.I.D.Z., MD 01092019-31082023), INTENSE LSH-NWO (C.I.D.Z., TTW/00798883), NWO Gravitation Program (C.I.D.Z., DBI2-001), NWO VIDI grant (Z.G., VI.Vidi.192.008), NWO-Klein grant (Z.G., OCENW.KLEIN.007) and ERC-stg grant (Z.G., 852869).

## Author contributions

Z.R. and Z.G. conceived the project. Z.R. performed most of the experiments and analysis. M.A. contributed to mice training and electrophysiology recording in the study. X.W. provided IpN electrophysiology data in DEC-trained mice and IpN microsimulation data. Z.R., C.I.D.Z., and Z.G. wrote the whole manuscript with inputs from all the authors. Z.G. supervised the whole project.

## Competing interests

The authors declare no competing interests.
