## [Transparent Peer Review file · Nature Communications]

Neuronal dynamics of cerebellum and medial prefrontal cortex in adaptive motor timing

Corresponding Author: Dr Zhenyu Gao

Version 0:

Reviewer comments:

Reviewer #1

(Remarks to the Author)

This manuscript investigated differential roles of the cerebral cortex and cerebellum during adaptive control of associative motor timing. This study found a very interesting behavior in which the onset timing of motion (CR) adapts almost instantaneously from DEC to TEC and vice versa. The authors recorded IpN and mPFC neurons during this paradigm switching. They found that Adaptation of CR timing is not fully explained by the changes of IpN modulation, and that mPFC neurons display adaptation of the CR-related modulations and baseline activities during the paradigm switching. Finally, they found that silencing of mPFC blocked the adaptation of CR timing. Although these findings contain potential impact and are interesting but not convincing due to the analysis and task design, which made their interpretation difficult, and logical gaps in their story.

Major comments

1. The analysis and task design.

They used CR onset as a measure of motor timing adaptation, but this measure is sensitive to the analysis method and makes interpretation difficult. They describe their detection method as "CR onset was defined as the timing with which eyelid closure exceeded three SDs of the baseline value." This is not appropriate. It is likely that CR onset is detected later than it actually is when the CR slope is small (e.g., when the CR amplitude itself is small); since the CR slopes are different for DEC and TEC, it is possible that the instantaneous adaptation after the paradigm switching reflects this problem. At the very least, they need to compare onsets with multiple definitions, such as the definition using Eyelid velocity. In addition, CR peak timing in US-omitted trials is a better indicator of movement timing in EBC. It is recommended to analyze this index as much as possible and compare it with onset timing.

The task design issue is particularly apparent in Fig 3, where Fig 3c is very similar to the Extinction data (peak timing has not changed at all). If CR (250ms and DEC) and CR (500ms and TEC) are independent components to some extent, we can assume that after the paradigm switching, the CR component (250ms and DEC) becomes smaller during Extinction and the CR component (500ms and TEC) becomes larger through learning. Considering that the learning of CR (500ms and TEC) is very slow (due to the long CS-US interval), this explains why the data in Fig. 3c is very similar to Extinction for CR (250ms and DEC). It would be interesting that the very fast CR onset shift happens in Extinction, but it is not the purpose of this paper, and the story would need to be modified significantly. I think it is necessary to compare how the CR onset changes with a simple CR (250 ms and DEC) Extinction in the experiment.

2. Regarding logical gaps

There are some major logical gaps scattered throughout the current manuscript. The most significant gap is the gap between Figs 8 and 9. This area is the most important part of this study, which examines the mechanism underlying the instantaneous adaptation of CR onsets. If the claim of the authors is correct, neurons in the mPFC should show the instantaneous adaptation. The changes in neuronal activity in Figure 8 all seem to change too slowly to be the basis for the instantaneous adaptation. If there are no underlying neurons, then the loss of adaptation of CRs in the inactivation experiment with muscimol (in Fig 9) cannot be interpreted as a result of inactivation of the basis of adaptation. In that case, other possibilities, such as the possibility that adaptation does not occur in CRs that have been reduced in size by muscimol, would be more plausible.

The second logical gap is that the network from the mPFC to the cerebellum via the pontine nucleus is the most important hypothesis for the interpretation of the results, but this seems unlikely. This hypothesis essentially assumes an information

flow from the mPFC to the cerebellum and then to the CR, which appears to contradict the authors' claims that the mPFC and CR show the instantaneous adaptation but that cerebellar nucleus IpN activity does not show the instantaneous adaptation. This major logical gap should be corrected, for example by proposing a different hypothesis.

Specific comments:

I don't think Delta facilitation (%) in Fig 1 is a good indicator. The delta firing rate (Hz) is better.

Fig. 1k is clearly not a normal distribution. Since non-parametric tests are used, Fig. 1k should be a non-parametric median + interquartile range.

In Fig 2, does the 100 ms from the facilitation onset to the CR onset not agree with previous reports? In particular, it does not agree with Fig 2 of their own study Brinke et al 2017 eLIFE. In Fig 2d,f, CR onset occurs at the end of the facilitation, which also looks totally different from Brinke et al 2017 eLIFE. I think it is necessary to reconsider the analysis.

The analysis of CR peak timing in Fig 2 should be done in CS-alone trials. If it is not possible, the panels should be excluded.

In Fig 3c and Fig 5c, CR dynamics should be presented according to trial (e.g., every 5 trials from the paradigm switching).

There are obvious visual responses in Fig 7. Shouldn't this be analyzed separately from subsequent activities?

In Fig 8, the time course of change in firing rate should be displayed for each population.

Reviewer #2

(Remarks to the Author)

This manuscript is thorough and reasonably complicated in presentation. It doesn't break new ground conceptually but documents some expected findings very thoroughly.

1. This manuscript explores behavioral and single unit functional shifts when mice are switched in training from delay to trace eyeblink conditioning, or vice versa. Michael Mauk and colleagues have done quite a few studies like this of behavior in rabbits. The novelty here is the single neuron recordings that were studied.
2. The fact that medial prefrontal cortex regions were required in order for the switches to occur, as shown in the muscimol inactivation experiments in Figure 9, was expected but interesting.
3. The electrodes used for single neuron recording were silicone probes, not usually used for chronic implants. Were the electrodes implanted chronically? If not, were they removed daily and reinserted on the following day? How many days were recordings done in each mouse; in each region?
4. When the DEC-TEC shift and especially when the TEC-DEC shift was done was extinction – or at least a reduction in conditioned behavior - observed?
5. In general, the behavioral changes observed during the paradigm shifts were modest. Although, since the behaviors were normalized to the maximum behavior to the maximum eyelid closure to a 1.2 microamp stimulus (line 505), we don't really know what the size of the behavioral change was.
6. Given that the behavioral changes were not large, it is not surprising that the single neuron firing rate changes during the shifts were also not large. This is certainly the case for the mPFC neuron changes during the shifts shown in Figure 8.
7. The authors don't say which subregion of mPFC they are recording from – they should make some statement on this point. The diagram shown in Figure 7 suggests prelimbic – but infralimbic region may also have been recorded from. This is important in attempting to interpret the firing rate changes observed.
8. Line 63 should cite the McCormick and Thompson 1984 paper.
9. Lines 106-108 state that the CR peak to US onset times were comparable between trace and delay conditioned mice. But Figure 1 D shows considerably shorter times in delay eyeblink conditioning.
10. Lines 477-478 indicate that an ITI of 8-12 s was used. Nordholm and Thompson (1991) showed in rabbit that a 9s ITI did not support "true" CRs. Does this indicate that mouse and rabbit eyeblink conditioning operates with different temporal requirements? Or what?

Reviewer #3

(Remarks to the Author)

This study by Ren et al. examines the activity/involvement of two brain regions, cerebellar IpN and cortical mPFC, during a flexible temporal adaptation task. Using eye blink conditioning, the authors train mice to learn either a short or long interval (so-called delay and trace eyeblink conditioning, which are believed to involve the cerebellum for the former and the cerebellum and cortex for the latter) and, after CR acquisition, the temporal contingency is switched. This manipulation allows identification how the behavior adapts and the neural correlates of the learned change. Surprisingly, the authors find that mice immediately adapt the timing of their CR response rather than taking many trials/sessions to learn the new timing. Using silicon probe recording, the authors find that IpN activity changes cannot explain the behavioral alteration. By contrast, mPFC showed alterations for each contingency implicating this region as an important locus in directing the behavioral change. To directly test the mPFC's involvement, the authors turn to pharmacological inactivation; the mice could no longer

acquire the temporal adaptation after musimol injection. Overall, this study adds a significant new understanding of neural dynamics underlying flexible timing and will certainly be of interest to many investigators. I have few concerns, mostly centered on methodological questions, that the authors should be able to easily address. These concerns do not diminish my overall high level of enthusiasm for this work.

- 1) Although DEC-trained mice rapidly adapt to the TEC interval; they do not reach the timing of TEC-trained animals indicated that some additional learning is required. Does the length of the TEC interval determine this performance level? i.e., if the interval was lengthened to 500ms or beyond, would DEC-trained mice show comparable, rapid adaptation?
- 2) Line 148: how are CR-related IpNs defined?
- 3) Line 155: please elaborate on "eyeblick-related" IpNs; this designation will be lost on readers unfamiliar with DEB
- 4) Line 168: there is no evidence that electrical stimulation only triggers the IpN-red nucleus pathway
- 5) Regarding encoding strategies for IpN that track CR onset (p7), were only facilitating neurons considered? If so, why not suppression cells? Does decoding accuracy change after adaptation to a change in timing?
- 6) Line 267: how are mPFC neurons with CR-related modulation defined?
- 7) That authors establish that mPFC activity is necessary in controlling CR temporal adaptation. However, a cohesive explanation of why is unclear. Is it keeping track of time or is it playing a more cognitive role such identifying that a task contingency has occurred and to apply a new strategy to compensate? A more pointed discussion would be helpful.
- 8) The manuscript has a substantial number of grammatical errors (way too many to list); it should be carefully edited as these errors detract from the quality of the presented work.

Version 1:

Reviewer comments:

Reviewer #1

(Remarks to the Author)

This study provides new insights into the cerebral and cerebellar mechanisms underlying the adaptive control of learned associative motor timing, and revealed that they play distinct roles. Furthermore, this study shows that cerebellar temporal information coding is more complex than previously thought, involving multiplex coding, and suggests that the paradigm-switch is signaled and controlled by the mPFC.

These findings offer a new perspective on our understanding of motor-timing control. The experiments and analyses are detailed, providing adequate evidence. The manuscript has significantly improved compared to the prior version, resulting in significantly fewer logical gaps and inconsistencies, both within the document itself and in comparison with previous studies. However, several concerns still remain, which are discussed below.

Major comments

The discussion has improved considerably, yet further improvements are needed. Given that the cerebellar nucleus can trigger the eyeblinks and its firing rate modulation starts early (>100 ms earlier than the actual CR onset, in TEC), the late CR onset in TEC requires suppression of cerebellar-triggered CR onset (the possibility that brain regions other than the cerebellum can initiate onset does not suffice). Since the authors indicated that the mPFC does not modify cerebellar activity in the shift of DEC-TEC, it's plausible to infer that the mPFC regulates onset timing by affecting the downstream pathway (red nucleus - facial nucleus - eyeblink muscles) rather than directly affecting the cerebellum. Therefore, discussion of the thalamo-cortical pathway is not sufficient, and the possibility of the downstream control needs to be discussed.

Confirming that the shift in CR onset is consistent regardless of the detection methods is a significant advance. However, as can be seen from the 10-20 ms delay from actual onset in Fig. 1d, the primary detection method lacks the detection sensitivity to accurately identify the onset. It seems that the most sensitive method (to the CR onset timing) among the three tested by the authors should be prioritized as the primary detection method.

Additionally, while CR onset is currently depicted only in Figures 1d and 2a, but it should be depicted in Figures 1i-j, 3b-d, 5b-d, as well. In these figures, the onset of CR for each individual trial, rather than just the session average, should be displayed (potentially in a Extended data Figure). Furthermore, given its significance, CR traces and onsets should also be included in Figure 9.

The observation of a shift in CR peak time in Extended Data Figures 5c and 9c is intriguing. However, the lack of division of epochs after adaptation into four parts makes comparison with the onset time course challenging. Dividing the post-adaptation epochs into four segments would enhance the analysis. Furthermore, Extended data Figure 5c seems to display two distinct groups of sessions, one with and one without peak delay. To clarify the differences between these two groups, it is necessary to include data of the CR traces of individual sessions.

Figures 3 and 5 are particularly important, yet they only present CR traces from a single session example. It's crucial to include figures that depict the CR traces and onsets from individual sessions, corresponding to Figures 3c and 5c. In this context, employing averages and error bars is not recommended. This approach could inaccurately skew the average CR onset due to the session with the shortest onset, potentially creating a misleading impression that the overall average onset is shorter than it actually is.

The manuscript describes the timing adaptation as being very rapid, occurring within a few trials, but no data supporting this claim are presented. A zoomed-in view of Figures 3f and 5f should be provided. This data is crucial as it demonstrates the time course of timing adaptation.

The influence of musimol injections on the mPFC during TEC appears to be less significant than previously documented. As outlined around line 284, mPFC activity is deemed essential for TEC, with prior research showing that musimol-induced mPFC inactivation led to the complete/severe loss of CRs in TEC. In contrast, this study observed only a decrease in CR probability without alterations to the other parameters. Clarifying whether the injection sites match those in prior studies is crucial for understanding these discrepancies.

Minor comment

The electrical stimulation of the cerebellar nuclei is conducted with 0.6-1.2 mA, which is considerably smaller compared to past studies. Please verify if this description is correct.

Reviewer #2

(Remarks to the Author)

The authors have responded appropriately to my comments.

I have no further suggestions on this manuscript.

The findings certainly are substantial enough to merit publication.

Reviewer #3

(Remarks to the Author)

The authors were attentive to my concerns, as well as those of the other reviewers. They addressed these each point-by-point, often with new experiments. Overall, this is an impressive body of work that will be useful to the field.

Version 2:

Reviewer comments:

Reviewer #1

(Remarks to the Author)

The authors provided sufficient data and proper responses to my comments. I have any more concerns or comments, except that trial 0 of Fig 5f should be DEC and be colored by orange (if trial 1 is the first DEC trial, the mouse should not change the CR onset timing because the mouse cannot predict the shift). This work is attractive to the field and suitable for publication.

Rebuttal letter

We thank the reviewers for their enthusiasm regarding our paper and for the constructive feedbacks that helped us to improve the paper. The reviewers raised important questions which led us to perform several major new experiments and analyses that further strengthened the key results.

1. Reviewers 1 and 2 asked about the differences between adaptation and extinction. In response to this, we performed a new extinction experiment in DEC-trained mice and clarified that the neural mechanisms behind rapid adaptation are distinct from those involved in extinction.
2. Reviewer 1 suggested using multiple methods for CR onset detection. We have performed detailed analysis using these detection methods and confirmed that different CR detection methods had little impact on the results.
3. Following the suggestion from reviewer 3 we designed a DEC-TEC750 adaptation paradigm, which consists of a 250 ms CS followed by a 500 ms CS-to-US interval. Our findings confirmed that mice were capable of adjusting the CR onset timing to align the optimal CR response with this novel paradigm.
4. Reviewer 1 and 3 requested a further discussion on the mPFC control of adaptive behaviour. We have elaborated our discussion on the possible functions and associated pathways. Additionally, we revised the relevant Fig.8 to better illustrate this notion, showing the instantaneous change of mPFC activity right after the paradigm change.
5. Finally, We have thoroughly revised the entire manuscript to improve its clarity and readability.

We hope that these revisions clarify the issues put forward by the reviewers. Please find a point-by-point rebuttal below.

Reviewer #1:

This manuscript investigated differential roles of the cerebral cortex and cerebellum during adaptive control of associative motor timing. This study found a very interesting behavior in which the onset timing of motion (CR) adapts almost instantaneously from DEC to TEC and vice versa. The authors recorded IpN and mPFC neurons during this paradigm switching. They found that Adaptation of CR timing is not fully explained by the changes of IpN modulation, and that mPFC neurons display adaptation of the CR-related modulations and baseline activities during the paradigm switching. Finally, they found that silencing of mPFC blocked the adaptation of CR timing. Although these findings contain potential impact and are interesting but not convincing due to the analysis and task design, which made their interpretation difficult, and logical gaps in their story.

We thank the reviewer for acknowledging the interesting points and potential impacts of our study and constructive suggestions. Regarding to the concerns, we have conducted several

new major experiments and more analysis following the suggestions of the reviewer. We think these new results further strengthen our original study.

Major comments

1. The analysis and task design.

They used CR onset as a measure of motor timing adaptation, but this measure is sensitive to the analysis method and makes interpretation difficult. They describe their detection method as "CR onset was defined as the timing with which eyelid closure exceeded three SDs of the baseline value." This is not appropriate. It is likely that CR onset is detected later than it actually is when the CR slope is small (e.g., when the CR amplitude itself is small); since the CR slopes are different for DEC and TEC, it is possible that the instantaneous adaptation after the paradigm switching reflects this problem. At the very least, they need to compare onsets with multiple definitions, such as the definition using Eyelid velocity. In addition, CR peak timing in US-omitted trials is a better indicator of movement timing in EBC. It is recommended to analyze this index as much as possible and compare it with onset timing.

We thank the reviewer for this insightful suggestion. Indeed, although detecting CR onset by standard deviation (SD) is an accepted method used in our previous work (Wang *et al* 2020¹), the precise timing may be influenced by the CR amplitude. Since the ways of capturing eyelid closure vary among different labs (e.g. EMG, magnetic sensor, camera), we think the SD criterion fits the best to our data with the least detection errors based on our previous studies¹. To further clarify this matter for this study, we followed the suggestion of the reviewer and analysed the CR onset timing by detecting the changes in CR velocity and the CR amplitudes, according to several previous studies^{2, 3}. We found that the comparison of CR onset timing between TEC and DEC were consistent by using three different detection methods. This also holds true for the DEC-to-TEC or TEC-to-DEC adaptation experiments. We have added these results to the new Extended data Fig. 1a-b, 5d-e, and 9d-e, together with new Fig.1e, Fig.3g, and Fig.5g. These results confirm that different CR kinematics in TEC and DEC have minimal influence on the detection of CR onset timing.

Moreover, we also analyzed the CR peak timing in the US-omitted probe trials during both DEC-to-TEC and TEC-to-DEC adaptation paradigms following the reviewer's suggestion. We found that not only the CR onset timing (new Fig.3g, and Fig.5g), but also the CR peak timing (new Extended data Fig. 5c, and 9c) were adapted in these paradigms. The difference in CR peak timing is still obvious between mice adapted to TEC and mice trained with TEC, which is in line with our findings by analyzing the CR onset (new Fig. 3g and Extended data Fig. 5c). All together our new analyses further support our initial claim that mice can flexibly adapt CS-US timings.

The task design issue is particularly apparent in Fig 3, where Fig 3c is very similar to the Extinction data (peak timing has not changed at all). If CR (250ms and DEC) and CR (500ms and TEC) are independent components to some extent, we can assume that after the paradigm switching, the CR component (250ms and DEC) becomes smaller during Extinction and the CR component (500ms and TEC) becomes larger through learning. Considering that the learning of CR (500ms and TEC) is very slow (due to the long CS-US interval), this explains why the data in Fig. 3c is very similar to Extinction for CR (250ms and DEC). It would be interesting that the very fast CR onset shift happens in Extinction, but it is not the purpose of

this paper, and the story would need to be modified significantly. I think it is necessary to compare how the CR onset changes with a simple CR (250 ms and DEC) Extinction in the experiment.

We acknowledge the reviewer's suggestions and agree that it's very important to clarify the neural mechanisms behind rapid adaptation and extinction. To address this question, we added a new set of experiments. We trained six mice using the DEC paradigm and subsequently performed extinction experiments, in which we observed a gradual decrease in CR amplitudes as previously described^{4, 5}. Interestingly, our detailed analysis, including trial-by-trial assessments, four quarterly segments, or whole-session averages (as shown in the new Figure 3h and the new Extended Data Figure 6), revealed that both the CR onset timing and peak timing remained largely unchanged during extinction. These findings highlight a clear distinction between the processes of adaptation and extinction. This suggests that when the unconditioned stimulus (US) is delayed, it plays a different role compared to when the US is omitted. Most interestingly, TEC-to-DEC adaptation resulted in, instead of a reduced CR amplitude, even an increased CR amplitude (new Extended Data Figure 9b). This provides compelling evidence against an extinction mechanism during adaptation, indicating that the adaptation process is distinct from a combination of extinction and re-learning. Furthermore, the data presented in the new Extended Data Figure 6b provides additional evidence that the timing of CR onset is not strongly correlated with the CR amplitude, confirming the analysis of CR onset timing from other behavioral parameters relevant to the previous question.

2. Regarding logical gaps

There are some major logical gaps scattered throughout the current manuscript. The most significant gap is the gap between Figs 8 and 9. This area is the most important part of this study, which examines the mechanism underlying the instantaneous adaptation of CR onsets. If the claim of the authors is correct, neurons in the mPFC should show the instantaneous adaptation. The changes in neuronal activity in Figure 8 all seem to change too slowly to be the basis for the instantaneous adaptation. If there are no underlying neurons, then the loss of adaptation of CRs in the inactivation experiment with muscimol (in Fig 9) cannot be interpreted as a result of inactivation of the basis of adaptation. In that case, other possibilities, such as the possibility that adaptation does not occur in CRs that have been reduced in size by muscimol, would be more plausible.

We apologize for the unclear illustration in Figs 8-9 of the previous version. We have revised Figure 8 to better emphasize the swift alteration in firing patterns immediately following the paradigm shift. In the updated Figure 8, we have divided the trials of adaptation epochs into four quarters and subsequently calculated the onset of CR and firing activity within each subgroup of trials. Our analysis revealed that the shift in firing patterns had already occurred during the first quarter of adaptation epochs, perfectly aligning with the onset of CR. Importantly, these observations held true for mPFC neurons recorded during both the DEC-to-TEC and TEC-to-DEC paradigms, as well as for both the modulation decrease and baseline increase subgroups. Therefore, our updated analyses confirm that the change in mPFC firing patterns during the adaptation task is not gradual but instantaneous. When considered alongside Figure 9, which presents both electrophysiological and pharmacological evidence, the critical role of the mPFC in rapid behavioral adaptation is further underscored.

The second logical gap is that the network from the mPFC to the cerebellum via the pontine nucleus is the most important hypothesis for the interpretation of the results, but this seems unlikely. This hypothesis essentially assumes an information flow from the mPFC to the cerebellum and then to the CR, which appears to contradict the authors' claims that the mPFC and CR show the instantaneous adaptation but that cerebellar nucleus IpN activity does not show the instantaneous adaptation. This major logical gap should be corrected, for example by proposing a different hypothesis.

This is an excellent question. We believe that the mPFC-pons-cerebellum pathway plays a vital role in the acquisition and maintenance of the eyeblink response, but it's less involved in rapid adaptation.

Previous studies (Wu et al⁶) highlighted the role of the mPFC-pons-cerebellum pathway in TEC. They demonstrated that the pons-projecting mPFC neurons became active during the stimulus-free epoch, providing a signal to bridge the time gap between CS and US. Our research aligns with these findings. We detected specific activities in the mPFC neurons during CR trials in both TEC and DEC-trained mice (refer to Extended data Fig. 11c-d). This suggests that during the epoch before but close to US, the mPFC neurons might encode CR-related information, which the cerebellum could then receive via the pons. To put it simply, it seems both mPFC and IpN neurons share a similar encoding strategy for CR: they exhibit stronger modulation in trials where CR amplitudes are higher (Extended data Fig. 3e-f for the cerebellum). This data transmission likely involves the mPFC-pons-cerebellum pathway.

Beyond CR-related data, mPFC neurons hold diverse information during TEC, DEC, and rapid adaptation processes that may not necessarily engage the pontine pathway. We identified a subset of mPFC neurons that register both the sensory-related signals timed with the CS onset and the CR-related signals. Furthermore, in some mPFC neurons we observed a swift alteration in their baseline firing rates during adaptation, a change not evident in cerebellum. This indicates that rapid adaptation signals related to shifts between TEC and DEC might not utilize the mPFC-pons-cerebellum pathway. Detailed downstream regions that control such adaptations still need closer scrutiny. We hinted at this theory in our original manuscript's concluding remarks. Apologies for any lack of clarity. We've taken steps to articulate our hypothesis more transparently in our updated version (lines 406-475).

Specific comments:

I don't think Delta facilitation (%) in Fig 1 is a good indicator. The delta firing rate (Hz) is better. Fig. 1k is clearly not a normal distribution. Since non-parametric tests are used, Fig. 1k should be a non-parametric median + interquartile range.

We have changed delta facilitation (%) to delta firing rate (Hz), with median + interquartile as the reviewer suggested in the new Fig.1k, n. The difference in delta firing rate is consistent with the difference in delta facilitation.

In Fig 2, does the 100 ms from the facilitation onset to the CR onset not agree with previous reports? In particular, it does not agree with Fig 2 of their own study Brinke et al 2017 eLIFE. In Fig 2d,f, CR onset occurs at the end of the facilitation, which also looks totally different from Brinke et al 2017 eLIFE. I think it is necessary to reconsider the analysis.

We employed slightly varied analytical approaches in these two studies. In the prior study³, we discerned the temporal relationship between the IpN firing rate and eyelid closure by identifying the time of peak cross-correlation. In contrast, for the current study, we computed the average difference between modulation onset timing and CR onset timing. It's worth noting that we opted for this approach because the temporal correlation method from ten Brinke et al. isn't ideal for contrasting onset timings during TEC-DEC and DEC-TEC adaptations. For precise calculation of peak temporal correlation, one must presume a fairly stable trial-by-trial relationship between IpN activity and eyelid closure, coupled with a sufficient number of trial repetitions. Given that these conditions weren't met during adaptation, the correlation coefficient r became highly variable. This variability was even more pronounced when analyzing individual quartiles, which had limited trial data. To maintain consistency in our current study, we preferred using the absolute difference over temporal correlation.

To address the reviewer's concerns further, we've analyzed our data using the methodology from the ten Brinke et al. 2017 study (as seen in the Rebuttal figure 1). Consistent with the previous study, the peak correlations between IpN timing and CR timing (the hotspot areas) are positioned above the diagonal line. This demonstrates that the IpN firing activity precedes the CR onset in both TEC- (a) and DEC- (b) trained mice. The peak correlation between firing rate and eyelid closure was approximately 50 ms for both sets (b, d), which doesn't statistically deviate from the previous study's results.

Rebuttal figure 1

Rebuttal figure 1: The temporal correlation between IpN activity and eyelid position in TEC- and DEC-trained mice. a Trial-by-trial temporal correlation heatmap between the facilitating IpN neurons and their corresponding eyelid positions during TEC. **b** The averaged

temporal correlation r value across different time intervals from the data showing in **a**. The timing of maximum correlation for each individual neuron are plotted above (Data is shown as median with interquartile range, $P < 0.0001$ comparing to 0 ms interval pair, $n = 138$ cells). **c** and **d** are the same as **a** and **b** but for DEC-trained mice (in **d** $P = 0.0025$, $n = 51$ cells).

The analysis of CR peak timing in Fig 2 should be done in CS-alone trials. If it is not possible, the panels should be excluded.

We agree. The panels related to peak time have been excluded from Fig. 2, and also from the extended data Fig. 2.

In Fig 3c and Fig 5c, CR dynamics should be presented according to trial (e.g., every 5 trials from the paradigm switching).

To better visualize the difference and match the following statistical analysis of CR dynamics during adaptation, we added one more panel plotting the average eyelid closure curves of the example sessions before paradigm change, together with those from the first and last 25% CR trials after the switches (see new Fig. 3c and Fig. 5c).

There are obvious visual responses in Fig 7. Shouldn't this be analyzed separately from subsequent activities?

We thank the reviewer for the suggestion. Indeed, we categorized mPFC neurons into transient and sustained types based on modulation duration in both TEC- and DEC-trained mice (Fig. 7d-g). For the entire transient group and some neurons in the sustained group, we observed modulation time fixed to the CS onset. However, we wondered if these were purely visual responses, given that the latency of these events sometimes exceeded 100 ms post-CS onset, much longer than an expected sensory input. We apologize for the lack of clarity. We have indeed separately analyzed the early peak and sustained activity, concluding that the sustained activities correlate with the CR (Extended data Fig.11c, d).

In Fig 8, the time course of change in firing rate should be displayed for each population.

We agree. We have updated the Fig. 8 to better illustrate this point (new Fig. 8d, f, j, and l).

Reviewer #2:

This manuscript is thorough and reasonably complicated in presentation. It doesn't break new ground conceptually but documents some expected findings very thoroughly.

1. This manuscript explores behavioral and single unit functional shifts when mice are switched in training from delay to trace eyeblink conditioning, or vice versa. Michael Mauk and colleagues have done quite a few studies like this of behavior in rabbits. The novelty here is the single neuron recordings that were studied.

2. The fact that medial prefrontal cortex regions were required in order for the switches to occur, as shown in the muscimol inactivation experiments in Figure 9, was expected but interesting.

We appreciate the reviewer's feedback on the thoroughness and innovative aspects of our study. Our results differ notably from the prior research by Michael Mauk and his team in multiple dimensions.

Firstly, regarding the experimental design: Mauk and his team explored an adaptation paradigm where rabbits initially trained with DEC (500ms CS) shifted to TEC (500ms CS plus 500ms interval), as detailed in Halverson et al., 2018. In their approach, most Purkinje cell recordings were conducted either before or after adaptation, with only two cells being recorded continuously from delay to trace conditioning. In contrast, our study trained mice to adapt in both directions. Moreover, as the reviewer rightly noted, we monitored the neural activity patterns of extensive neuron populations in both the cerebellar nuclei and mPFC throughout the adaptation process.

Secondly, the primary outcomes from our experiments diverge considerably from the rabbit-based studies. In rabbits, adapting from DEC to TEC requires over a hundred training trials. During this period, the CR response to DEC progressively fades, while a new CR response to TEC arises after continuous training. This gradual adaptation is mirrored in the two Purkinje cells recorded by Mauk's team during the process, where the CR modulation waned and then re-emerged during post-adaptation. In our study, we unveiled a unique mechanism governing swift adaptations between DEC-TEC and TEC-DEC within several trials in mice. We followed neural modulations in both the cerebellar nuclei and mPFC. Interestingly, modulation in the cerebellar nuclei didn't faithfully correspond with the observed adaptation, but we identified clear patterns in mPFC neurons linked with adaptation.

Lastly, our conclusions don't fully concur with Halverson et al.'s findings. Their study suggested that changes related to learning in the cerebellum regulated the timing and magnitude of cerebellar responses consistently across both DEC and TEC paradigms. In contrast, we identified considerable variations in the coding strategies for DEC and TEC among the cerebellar nuclei neurons. Moreover, our findings highlight a cerebellum-independent but mPFC-dependent mechanism for rapid adaptation. Consequently, we suggest that this rapid adaptation triggers extracerebellar circuits that differ from those underpinning cerebellar learning.

Hence, we believe our study significantly advances our understanding of how population activity in the frontal cortex and cerebellum influences the flexible adaptive control of sensorimotor timing.

3. The electrodes used for single neuron recording were silicone probes, not usually used for chronic implants. Were the electrodes implanted chronically? If not, were they removed daily and reinserted on the following day? How many days were recordings done in each mouse; in each region?

We performed acute recordings throughout the study. The 64-channel silicon probes were inserted and removed daily. In general, we recorded 5 days for each recording regions. Except for the TEC-to-DEC adaptation paradigms, we recorded for the first two days to avoid long term adaptation due to repetitive training. We have added this detail in the method section.

4. When the DEC-TEC shift and especially when the TEC-DEC shift was done was extinction – or at least a reduction in conditioned behavior - observed?

For DEC-to-TEC adaptation, we observed a decreased CR amplitude during adaptation. To distinguish adaptation from extinction, we performed a new set of experiment in which mice with DEC followed extinction. As indicated in the new Fig. 3h and Extended Data Fig. 6, while CR amplitudes were gradually reduced during extinction, the CR onset and peak timing remained constant. This experiment excludes the possibility that shifting in CR timing during DEC-to-TEC was induced by CR extinction. Moreover, the CR percentage remains unchanged after adaptation (Extended Data Fig. 5a), further confirming that DEC-to-TEC adaptation has distinct features from extinction.

Most interestingly, the TEC-to-DEC adaptation, instead of extinction, induced even an increased CR amplitude after adaptation (Fig. 5). This provides compelling evidence against an extinction mechanism during adaptation, indicating that the adaptation process is distinct from a combination of extinction and re-learning.

5. In general, the behavioral changes observed during the paradigm shifts were modest. Although, since the behaviors were normalized to the maximum behavior to the maximum eyelid closure to a 1.2 microamp stimulus (line 505), we don't really know what the size of the behavioral change was.

We apologize for the lack of clarity. The electric stimulation was exclusively utilized in a specific segment of our experiment (Extended Data Fig. 3) to determine the minimal delay between IpN activation and eyelid closure. In other experiments, we determined the maximum eyelid closure using the peak amplitudes of the airpuff-induced unconditional responses, which typically correspond to complete eyelid closure. Given this context, we assert that the observed behavioral changes were substantial. For instance, during the TEC-to-DEC adaptation, there was a shift in CR onset timing by more than 50ms post-adaptation (Fig. 5e, g). Furthermore, the CR amplitudes rose from 22% to 65% of full eyelid closure by the end of the CS duration (Fig. 5c).

6. Given that the behavioral changes were not large, it is not surprising that the single neuron firing rate changes during the shifts were also not large. This is certainly the case for the mPFC neuron changes during the shifts shown in Figure 8.

As highlighted above, we noted pronounced alterations at the behavioral level during DEC-TEC and TEC-DEC adaptation. Simultaneously, mPFC neurons exhibited significant adaptation both at individual and population levels. To convey these shifts more effectively, we have restructured Fig. 8. For example, a particular mPFC neuron displayed a pronounced reduction in its modulation following DEC-to-TEC adaptation (Fig. 8c), while another exhibited the contrary pattern (Fig. 8i).

7. The authors don't say which subregion of mPFC they are recording from – they should make some statement on this point. The diagram shown in Figure 7 suggests prelimbic – but infralimbic region may also have been recorded from. This is important in attempting to interpret the firing rate changes observed.

We agree and we have highlighted the recording region, which is indeed the prelimbic area, in the revised version. However, we can not exclude the possibility that a minor fraction of neurons was recorded from the infralimbic as the tip of recording tracks were not always clearly visible.

8. Line 63 should cite the McCormick and Thompson 1984 paper.

This paper has been cited in our new manuscript.

9. Lines 106-108 state that the CR peak to US onset times were comparable between trace and delay conditioned mice. But Figure 1 D shows considerably shorter times in delay eyeblink conditioning.

We are sorry about this confusion. For our analysis of the CR peak to US onset times, we used exclusively the probe trials (US omitted trials), but not the CS-US paired trials depicted in Fig. 1d. This approach was taken as the unconditioned response could potentially obscure the true peak of the CR. The CR peak times of the probe trials from TEC- or DEC-trained mice were comparable, see Rebuttal figure 2. We have expanded our explanations in this section to clarify this aspect.

Rebuttal figure 2

Rebuttal figure 2: The CR peak timings of the probe trials from TEC- and DEC-trained mice. Time zero indicates the timing when US was supposed to occur in paired trials ($P = 0.08$, $n = 92$ and 27 sessions).

10. Lines 477-478 indicate that an ITI of 8-12 s was used. Nordholm and Thompson (1991) showed in rabbit that a 9s ITI did not support "true" CRs. Does this indicate that mouse and rabbit eyeblink conditioning operates with different temporal requirements? Or what?

Indeed, this may suggest a difference in their temporal requirements across different species. The inter-trial-intervals from the studies using rabbits were typically about 30 s^{7, 8, 9}, whereas studies using mouse as a model use inter-trial-intervals from 5 s to 20 s^{1, 2, 3, 10, 11, 12}. Our study is therefore consistent with our previous study^{3, 10} and previous literature.

Reviewer #3:

This study by Ren et al. examines the activity/involvement of two brain regions, cerebellar IpN and cortical mPFC, during a flexible temporal adaptation task. Using eye blink conditioning, the authors train mice to learn either a short or long interval (so-called delay and trace eyeblink conditioning, which are believed to involve the cerebellum for the former and the cerebellum and cortex for the latter) and, after CR acquisition, the temporal contingency is switched. This manipulation allows identification how the behavior adapts and the neural correlates of the learned change. Surprisingly, the authors find that mice immediately adapt the timing of their CR response rather than taking many trials/sessions to learn the new timing. Using silicon probe recording, the authors find that IpN activity changes cannot explain the behavioral alteration. By contrast, mPFC showed alterations for each contingency implicating this region as an important locus in directing the behavioral change. To directly test the mPFC's involvement, the authors turn to pharmacological inactivation; the mice could no longer acquire the temporal adaptation after musimol injection. Overall, this study adds a significant new understanding of neural dynamics underlying flexible timing and will certainly be of interest to many investigators. I have few concerns, mostly centered on methodological questions, that the authors should be able to easily address. These concerns do not diminish my overall high level of enthusiasm for this work.

We are grateful to the positive feedback and high enthusiasm of the reviewer.

1) Although DEC-trained mice rapidly adapt to the TEC interval; they do not reach the timing of TEC-trained animals indicated that some additional learning is required. Does the length of the TEC interval determine this performance level? i.e., if the interval was lengthened to 500ms or beyond, would DEC-trained mice show comparable, rapid adaptation?

We thank the reviewer for this interesting question. In response to the reviewer's suggestion, we conducted set of major experiment. Specifically, we subjected mice to DEC training and implemented a novel adaptation paradigm TEC750, which consists of a 250 ms CS followed by a 500 ms CS-to-US interval (new Extended Data Fig. 7a). Consistently, our findings indicated that mice were capable of adjusting the CR onset timing to align with this novel paradigm (see new Extended Data Fig. 7b and c). Notably, the extended CS-to-US interval posed a greater challenge to the mice, resulting in a slower adaptation rate when compared to the TEC500 paradigm.

2) Line 148: how are CR-related IpNs defined?

We define CR-related IpNs as neurons that exhibit spike rate changes exceeding three times the standard deviation of the baseline frequency. This definition has been incorporated into the 'electrophysiology analysis' section of the methods.

3) Line 155: please elaborate on "eyeblink-related" IpNs; this designation will be lost on readers unfamiliar with DEB

We agree, a detailed description has been added to this part.

4) Line 168: there is no evidence that electrical stimulation only triggers the IpN-red nucleus pathway

We agree that electrical stimulation of the IpN-red nucleus pathway may trigger other movements. We have revised our statement to avoid over stating.

5) Regarding encoding strategies for IpN that track CR onset (p7), were only facilitating neurons considered? If so, why not suppression cells? Does decoding accuracy change after adaptation to a change in timing?

In our decoding analysis, we focused exclusively on the facilitating neurons, as the majority of neurons exhibited facilitation. There were insufficient numbers of suppression neurons to conduct a reliable population-level analysis. Also, decoding DEC trials posed a significant technical challenge for our decoder. Due to the low variability in CR timing during DEC trials, our decoder was unable to generate reliable predictions of CR timing. This is certainly an interesting direction to explore in future studies.

6) Line 267: how are mPFC neurons with CR-related modulation defined?

An mPFC neuron exhibiting CR-related modulation was defined as a neuron with a statistically significant increase in firing rate during the CS-US interval compared to its baseline. We have clarified this definition in the 'electrophysiology analysis' part of the method section.

7) That authors establish that mPFC activity is necessary in controlling CR temporal adaptation. However, a cohesive explanation of why is unclear. Is it keeping track of time or is it playing a more cognitive role such identifying that a task contingency has occurred and to apply a new strategy to compensate? A more pointed discussion would be helpful.

This is an excellent question. We think mPFC neurons are involved in both keeping track of time and signaling the switching of task contingency. Prior studies, such as Wu et al⁶, have highlighted the role of the mPFC-pons-cerebellum pathway in TEC. They demonstrated that the pons-projecting mPFC neurons become active during the stimulus-free epoch, bridging the time between CS and US. Our findings certainly aligns with this hypothesis. We detected activities in the mPFC neurons during CR trials in TEC-trained mice (Extended data Fig. 11c-d). This suggests that during the stimulus-free epoch, the mPFC neurons might encode CR-related information, which is relayed to the cerebellum via the pons.

Beyond CR-related information, mPFC neurons may hold diverse information during rapid adaptation processes that may not necessarily engage the pontine pathway. We identified a subset of mPFC neurons that register both the sensory-related signals timed with the CS onset and the CR-outcome related signals. Furthermore, in some mPFC neurons we observed a swift alteration in their baseline firing rates during adaptation—a change not evident in the cerebellum. This indicates that signals related to the shifts of task contingency are encoded in mPFC. Detailed downstream regions that control such adaptations still need closer scrutiny. We hinted at this theory in our original manuscript's concluding remarks. We have now discussed this point in more detail (lines 406-475).

8) The manuscript has a substantial number of grammatical errors (way too many to list); it should be carefully edited as these errors detract from the quality of the presented work.

We have thoroughly revised the entire manuscript to improve its clarity and readability.

References

1. Wang X, Yu SY, Ren Z, De Zeeuw CI, Gao Z. A FN-MdV pathway and its role in cerebellar multimodular control of sensorimotor behavior. *Nature communications* **11**, 6050 (2020).
2. Ohmae S, Medina JF. Climbing fibers encode a temporal-difference prediction error during cerebellar learning in mice. *Nature neuroscience* **18**, 1798-1803 (2015).
3. Ten Brinke MM, *et al.* Dynamic modulation of activity in cerebellar nuclei neurons during pavlovian eyeblink conditioning in mice. *eLife* **6**, (2017).
4. Medina JF, Nores WL, Mauk MD. Inhibition of climbing fibres is a signal for the extinction of conditioned eyelid responses. *Nature* **416**, 330-333 (2002).
5. Hu C, Zhang L-B, Chen H, Xiong Y, Hu B. Neurosubstrates and mechanisms underlying the extinction of associative motor memory. *Neurobiology of Learning and Memory* **126**, 78-86 (2015).
6. Wu GY, *et al.* Optogenetic Inhibition of Medial Prefrontal Cortex-Pontine Nuclei Projections During the Stimulus-free Trace Interval Impairs Temporal Associative Motor Learning. *Cerebral cortex* **28**, 3753-3763 (2018).
7. Halverson HE, Khilkevich A, Mauk MD. Cerebellar Processing Common to Delay and Trace Eyelid Conditioning. *The Journal of neuroscience : the official journal of the Society for Neuroscience* **38**, 7221-7236 (2018).
8. Weible AP, Weiss C, Disterhoft JF. Activity profiles of single neurons in caudal anterior cingulate cortex during trace eyeblink conditioning in the rabbit. *Journal of neurophysiology* **90**, 599-612 (2003).
9. Siegel JJ, Kalmbach B, Chitwood RA, Mauk MD. Persistent activity in a cortical-to-subcortical circuit: bridging the temporal gap in trace eyelid conditioning. *Journal of neurophysiology* **107**, 50-64 (2012).
10. ten Brinke MM, *et al.* Evolving Models of Pavlovian Conditioning: Cerebellar Cortical Dynamics in Awake Behaving Mice. *Cell reports* **13**, 1977-1988 (2015).
11. Heiney SA, Wohl MP, Chettih SN, Ruffolo LI, Medina JF. Cerebellar-dependent expression of motor learning during eyeblink conditioning in head-fixed mice. *The Journal of neuroscience : the official journal of the Society for Neuroscience* **34**, 14845-14853 (2014).
12. Albergaria C, Silva NT, Pritchett DL, Carey MR. Locomotor activity modulates associative learning in mouse cerebellum. *Nature neuroscience* **21**, 725-735 (2018).

Rebuttal letter

We thank the reviewers for their endorsements of our paper and for providing further suggestions. We have revised several figures in the manuscript accordingly to enhance the clarity. Additionally, we have expanded our discussion on the potential downstream pathways involved in the control of CR timing during adaptation. We hope these revisions adequately address the concerns raised by the reviewer. Below is our point-by-point rebuttal.

Reviewer 1:

This study provides new insights into the cerebral and cerebellar mechanisms underlying the adaptive control of learned associative motor timing, and revealed that they play distinct roles. Furthermore, this study shows that cerebellar temporal information coding is more complex than previously thought, involving multiplex coding, and suggests that the paradigm-switch is signaled and controlled by the mPFC.

These findings offer a new perspective on our understanding of motor-timing control. The experiments and analyses are detailed, providing adequate evidence. The manuscript has significantly improved compared to the prior version, resulting in significantly fewer logical gaps and inconsistencies, both within the document itself and in comparison with previous studies. However, several concerns still remain, which are discussed below.

Thank you for reviewing the updated version of the manuscript and acknowledging our study. In response to the concerns raised, we have conducted further analysis and expanded the discussion following your suggestions. We think these changes further clarify the information presented throughout the manuscript and strengthen our findings.

Major comments

The discussion has improved considerably, yet further improvements are needed. Given that the cerebellar nucleus can trigger the eyeblinks and its firing rate modulation starts early (>100 ms earlier than the actual CR onset, in TEC), the late CR onset in TEC requires suppression of cerebellar-triggered CR onset (the possibility that brain regions other than the cerebellum can initiate onset does not suffice). Since the authors indicated that the mPFC does not modify cerebellar activity in the shift of DEC-TEC, it's plausible to infer that the mPFC regulates onset timing by affecting the downstream pathway (red nucleus - facial nucleus - eyeblink muscles) rather than directly affecting the cerebellum. Therefore, discussion of the thalamo-cortical pathway is not sufficient, and the possibility of the downstream control needs to be discussed.

We agree with the view that mPFC could regulate CR timing by affecting other pathways downstream to the cerebellum. Indeed, the roles of mPFC in controlling timing have been reported in several studies. In the revised manuscript (Lines 458-483), we have added additional discussion on the potential role of mPFC-basal ganglia pathway, which is known for its key role in controlling movement timing. In short, the premotor nucleus of the eyelid muscle, the red nucleus, receives considerable projections from both the cortico-thalamic and cortico-basal ganglia pathways. The integrative regulation of the cerebral and cerebellar convergence

onto the red nucleus neurons could potentially facilitate the temporal control of adaptive CR downstream to the cerebellar outputs.

Confirming that the shift in CR onset is consistent regardless of the detection methods is a significant advance. However, as can be seen from the 10-20 ms delay from actual onset in Fig. 1d, the primary detection method lacks the detection sensitivity to accurately identify the onset. It seems that the most sensitive method (to the CR onset timing) among the three tested by the authors should be prioritized as the primary detection method.

Additionally, while CR onset is currently depicted only in Figures 1d and 2a, but it should be depicted in Figures 1i-j, 3b-d, 5b-d, as well. In these figures, the onset of CR for each individual trial, rather than just the session average, should be displayed (potentially in a Extended data Figure). Furthermore, given its significance, CR traces and onsets should also be included in Figure 9.

We apologize for the confusing illustration of CR onset timing. Indeed, the 10-20 ms delay from the actual onset in Fig. 1d is because we indicated the onset timing of the session averaged trace, instead of individual trials. To alleviate the concern regarding the detection method, we further compared the outcomes of CR onset timings using three detection methods adopted in the previous revision experiments: above 6x standard deviation (SD), 5% of the amplitude, and 0.2% CR velocity. As shown in Rebuttal Figure 1, these detection methods resulted in no statistical significance in CR onset timings. Hence, our current method offers sufficient accuracy and sensitivity for detecting CR onset.

Furthermore, we have now plotted the CR onset timings of individual trials, as well as the session averages in Figures 1d, 3b,c, 5b,c, and 9c,e. The modified illustration provides further clarity to the figures.

Rebuttal figure 1

Figure 1

Rebuttal figure 1: The comparison of three CR onset detection methods.

a The average CR onset was detected using three methods (see Methods): method-1, 6×standard deviation (SD); method-2, 5% of CR amplitude (Amplitude); and method-3, above 0.2% of full eyelid closure per millisecond (Velocity) for all the sessions under DEC paradigm ($n = 135$ sessions). **b** Same as (a) but for all the sessions under TEC paradigm ($n = 161$ sessions). Data are shown as the mean \pm s.e.m.

The observation of a shift in CR peak time in Extended Data Figures 5c and 9c is intriguing. However, the lack of division of epochs after adaptation into four parts makes comparison with the onset time course challenging. Dividing the post-adaptation epochs into four segments would enhance the analysis. Furthermore, Extended data Figure 5c seems to display two distinct groups of sessions, one with and one without peak delay. To clarify the differences between these two groups, it is necessary to include data of the CR traces of individual sessions.

Thank you for the suggestion on dividing the post-adaptation epochs into four segments. The new plots in Extended Data Fig. 5d (DEC-to-TEC adaptation) and Fig. 9c (TEC-to-DEC adaptation) indeed show that the CR peak time rapidly adapted and then stabilized after the paradigm switch. This presentation enhances the analysis and further supports our conclusion that rapid adaptation occurs in CR timing.

For the second comment regarding to previous Extended data Fig.5c (now is Extended data Fig.5b), we now calculated the changes in CR peak time of each paradigm switch session and plotted the distribution histogram of these changes. As shown in the new Extended data Fig.5c, the majority, 29 out of 38 (76.3%), sessions showed delayed probe trial CR peak time after DEC-to-TEC adaptation. Several sessions did not show detectable change; however the overall distribution appears to be widespread rather than two distinct groups.

Figures 3 and 5 are particularly important, yet they only present CR traces from a single session example. It's crucial to include figures that depict the CR traces and onsets from individual sessions, corresponding to Figures 3c and 5c. In this context, employing averages and error bars is not recommended. This approach could inaccurately skew the average CR onset due to the session with the shortest onset, potentially creating a misleading impression that the overall average onset is shorter than it actually is.

We agree. We now incorporate CR traces and onset distribution in the new Fig. 3d and 5d. These new figures depict the CR onsets of individual trials during different stages of adaptation.

The manuscript describes the timing adaptation as being very rapid, occurring within a few trials, but no data supporting this claim are presented. A zoomed-in view of Figures 3f and 5f should be provided. This data is crucial as it demonstrates the time course of timing adaptation.

We agree. Zoomed-in views of Fig. 3f and 5f, focusing on the 10 trials before and 10 trials after the adaptation are now added.

The influence of musimol injections on the mPFC during TEC appears to be less significant than previously documented. As outlined around line 284, mPFC activity is deemed essential for TEC, with prior research showing that musimol-induced mPFC inactivation led to the complete/severer loss of CRs in TEC. In contrast, this study observed only a decrease in CR probability without alterations to the other parameters. Clarifying whether the injection sites match those in prior studies is crucial for understanding these discrepancies.

Thank you for highlighting this intriguing point. We agree that the injection sites and injection volume could potentially account for the observed differences. The mPFC is a very elongated structure, our injections were predominantly in the rostral-medial prefrontal area. Previous work by Takehara-Nishiuchi et al.¹ reported that muscimol (1 μ l, 1 mM) inhibition of the mPFC (prefrontal area) affected TEC acquisition but not DEC in rats. In contrast, Wu et al.² infused the same amount of muscimol into the caudal mPFC in rats, showing that blocking mPFC did not affect CR when CS was sufficiently strong. Siegel et al.³ injected muscimol (100 nl, 1 mM) in TEC-trained mice, with injection sites more posterior and dorsal than in our study. Mice had impaired CR percentage and amplitude. In our study we injected typically less than 20 nl of muscimol, likely resulting in much less spread within the mPFC than in all the other studies. We observed a clear reduction in the CR percentage (see Extended Data Figure 12), but not the CR amplitudes. Therefore, we speculate that both the specific injection region and/or the extent of muscimol coverage in mPFC could contribute to the differential effects on TEC.

Minor comment

The electrical stimulation of the cerebellar nuclei is conducted with 0.6-1.2 mA, which is considerably smaller compared to past studies. Please verify if this description is correct.

We have confirmed the stimulation protocols. In a closely related study, 1-15 μ A stimulations in the InP were used to evoke eyelid closure⁴. We attribute this difference to variations in the microstimulation electrodes. Specifically, the study employed an 80- μ m-diameter platinum iridium monopolar electrode, whereas we used a glass electrode with an 8- μ m-diameter tip opening filled with 2M saline. This might contribute to the difference in current intensities we used.

Reviewer 2:

The authors have responded appropriately to my comments.

I have no further suggestions on this manuscript.

The findings certainly are substantial enough to merit publication.

Thank you for your endorsement of the study.

Reviewer 3:

The authors were attentive to my concerns, as well as those of the other reviewers. They addressed these each point-by-point, often with new experiments. Overall, this is an impressive body of work that will be useful to the field.

Thank you for your endorsement of the study.

References

1. Takehara-Nishiuchi K, Kawahara S, Kirino Y. NMDA receptor-dependent processes in the medial prefrontal cortex are important for acquisition and the early stage of consolidation during trace, but not delay eyeblink conditioning. *Learn Mem* **12**, 606-614 (2005).
2. Wu GY, *et al.* Medial Prefrontal Cortex-Pontine Nuclei Projections Modulate Suboptimal Cue-Induced Associative Motor Learning. *Cerebral cortex* **28**, 880-893 (2018).
3. Siegel JJ, *et al.* Trace Eyeblink Conditioning in Mice Is Dependent upon the Dorsal Medial Prefrontal Cortex, Cerebellum, and Amygdala: Behavioral Characterization and Functional Circuitry. *eNeuro* **2**, (2015).
4. Heiney SA, Wohl MP, Chettih SN, Ruffolo LI, Medina JF. Cerebellar-dependent expression of motor learning during eyeblink conditioning in head-fixed mice. *The Journal of neuroscience : the official journal of the Society for Neuroscience* **34**, 14845-14853 (2014).

Reviewer #1 (Remarks to the Author):

The authors provided sufficient data and proper responses to my comments. I have any more concerns or comments, except that trial 0 of Fig 5f should be DEC and be colored by orange (if trial 1 is the first DEC trial, the mouse should not change the CR onset timing because the mouse cannot predict the shift). This work is attractive to the field and suitable for publication.

We thank the reviewer for acknowledging our work. Regarding the point about Fig. 5f, we agree and have changed the color code of trial 1 to orange, which represents the DEC epoch.